# Adaptive Self-Distillation for Minimizing Client Drift in Heterogeneous Federated Learning

**M. Yashwanth**                                              *yashwanthm@iisc.ac.in*
*Indian Institute of Science*

**Gaurav Kumar Nayak**                              *gauravkumar.nayak@mfs.iitr.ac.in*
*Indian Institute of Technology (IIT) Roorkee*

**Arya Singh**                                      *f20180762g@alumni.bits-pilani.ac.in*
*Indian Institute of Science*

**Yogesh Simmhan**                                            *simmhan@iisc.ac.in*
*Indian Institute of Science*

**Anirban Chakraborty**                                        *anirban@iisc.ac.in*
*Indian Institute of Science*

**Reviewed on OpenReview:** *https://openreview.net/forum?id=K58n87DE4s*

## Abstract

Federated Learning (FL) is a machine learning paradigm that enables clients to jointly train a global model by aggregating the locally trained models without sharing any local training data. In practice, there can often be substantial heterogeneity (e.g., class imbalance) across the local data distributions observed by each of these clients. Under such non-iid label distributions across clients, FL suffers from the 'client-drift' problem where every client drifts to its own local optimum. This results in slower convergence and poor performance of the aggregated model. To address this limitation, we propose a novel regularization technique based on adaptive self-distillation (ASD) for training models on the client side. Our regularization scheme adaptively adjusts to each client's training data based on the global model's prediction entropy and the client-data label distribution. We show in this paper that our proposed regularization (ASD) can be easily integrated atop existing, state-of-the-art FL algorithms, leading to a further boost in the performance of these off-the-shelf methods. We theoretically explain how incorporation of ASD regularizer leads to reduction in client-drift and empirically justify the generalization ability of the trained model. We demonstrate the efficacy of our approach through extensive experiments on multiple real-world benchmarks and show substantial gains in performance when the proposed regularizer is combined with popular FL methods. The link to the code is `https://github.com/vcl-iisc/fed-adaptive-self-distillation`.

## 1 Introduction

Federated Learning (FL) is a machine learning paradigm where the clients collaboratively learn a shared model under the orchestration of the server without sharing any of their local training data with other clients or the server. Due to the privacy-preserving nature of FL, it has found many applications in smartphones (Hard et al., 2018; Ramaswamy et al., 2019), the Internet of Things (IoT), healthcare organizations (Rieke et al., 2020; Xu et al., 2021), where training data is generated at edge devices or from privacy-sensitive domains. As originally introduced in (McMahan et al., 2017), FL involves model training across an architecture consisting

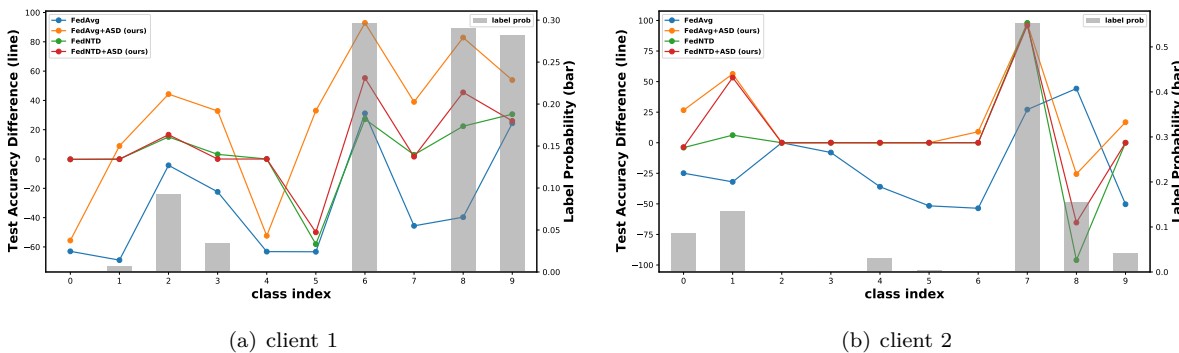

(a) client 1        (b) client 2

Figure 1: Impact of one round of local training on the test accuracy of two clients with different label distribution sampled from CIFAR-10 dataset: The effect of local learning on test accuracy is analyzed by measuring the change in accuracy before and after local training, with positive values indicating improved model performance. Interestingly, in scenarios where classes with low probability of occurrence or under-represented, models trained using FedAvg frequently exhibit a decline in accuracy post-training. In contrast, incorporating our proposed adaptive self-distillation regularizer (ASD) into FedAvg (FedAvg+ASD) not only effectively captures knowledge from well-represented classes but also preserves information about under-represented classes. A similar pattern is observed with FedNTD and FedNTD+ASD.

of one server and multiple clients. In traditional FL, each client securely holds its training data due to privacy concerns as well as to avoid large communication overheads while transmitting the same. At the same time, these clients aim to collaboratively train a generalized model that can leverage the entirety of the training data disjointly distributed across clients.

Data ingested at the edge/client devices are often highly heterogeneous as a consequence of the data generation process. They can differ in terms of quantity imbalance (the number of samples at each client are different), label imbalance (empirical label distribution across the clients widely vary), and feature imbalance (features of the data across the clients are non-iid). When there exists a label or feature imbalance, the objective for every client becomes different as the local minimum for every client objective will be different. In such settings, during the local training, the client's model starts to drift towards its local minimum and farther away from the global objective. This is undesirable as the goal of FL is to converge to a global model that generalizes well across all the clients. This phenomenon, known as 'client-drift', is introduced and explored in earlier works (Karimireddy et al., 2020; Acar et al., 2021; Wang et al., 2021). In this work, we will be considering only the label heterogeneity. In any given FL round, the client initializes its model with global model weights and then starts training its model using the local data. Due to this, the client training often leads to overfitting the local data and cannot retain the knowledge acquired from the global model in an earlier FL round.

Recently (He et al., 2022b) introduced a class-wise adaptive weighting scheme (FedCAD) at the server side. The major drawback of FedCAD is that it assumes the presence of related auxiliary data and reliability on the server to compute the weights for the clients. Dependency on the server for computing the adaptive class-wise weights necessitates the availability of auxiliary data at the server. Another work (Lee et al., 2022) proposes FedNTD which poses the client-drift as a local forgetting problem. It cannot mitigate client-drift effectively since it assigns uniform weights to regularization loss for all samples, independent of label distribution. Consequently, it treats high and low probability samples similarly, biasing the client model towards those with higher probability of occurrence, thus degrading performance. To address these issues and motivated by client model regularization in mitigating client drift and to remove the server's dependency on computing client-side weights, we introduce a computationally efficient strategy known as Adaptive Self-Distillation (ASD) for Federated Learning. Importantly, ASD does not require any auxiliary data. We use the KL divergence between the global and local models as the regularizer. For every sample, the weight assigned to the regularization loss is adaptively adjusted based on the global model's prediction entropy and the

empirical label distribution of the client's data. Specifically, when the server model encounters samples with high entropy, we reduce the weighting on the regularization loss, whereas, for samples with a low probability of occurrence, we prioritize the learning from the global model. This adaptive approach enables local models to effectively learn from the cross-entropy loss for more frequent labels while leveraging the global model's guidance for less frequent labels. The adaptive weights are computed without relying on external or proxy data, unlike methods such as FedCAD which relies on external data. Moreover, the additional computational burden on clients is minimal, involving only a single forward pass of the training data.

In Fig 1, we explain how the ASD regularization with adaptive weights helps mitigate the client-drift. We analyze the impact of client-drift by observing one round of local training on a particular client model with the CIFAR-10 dataset. We see that FedAvg substantially deteriorates the performance on the labels that have sparse or no representation in the client's local data. After adding the ASD loss, the impact is reduced. The ASD with (adaptive weights) performs the best in terms of local learning and preserving the global model knowledge on the sparse classes. We theoretically explain the client-drift reduction through our proposed ASD regularizer. In addition, we also provide justification on how ASD leads to improved generalization of the global model. This novel design of our proposed method allows the regularizer to be easily integrated atop any existing FL methods, and this results in substantial performance gains, making it an attractive and compelling solution to the federated learning problem. To the best of our knowledge, this is the first work where the adaptive weights are used for the distillation loss in the FL framework without requiring access to auxiliary data and without the assistance of the server. We would like to clearly point out that the goal of this work is not to directly compete against any particular regularization method used in FL. Our proposed ASD regularizer is of a true plug-and-play nature. With a very negligible computational overhead (discussed in Sec. 6), ASD can be used as an additional regularization on top of any off-the-shelf FL method (either with regularized FL methods such as FedProx, FedDyn or FL methods without regularization such as FedSAM, FedAvg etc.) and further boost their performance across the benchmark datasets such as CIFAR-100/10 and Tiny-ImageNet for both IID and non-IID settings. As a validation, we combine our proposed method with some of the popular off-the-shelf FL methods such as FedAvg (McMahan et al., 2017), FedProx (Li et al., 2020), FedDyn (Acar et al., 2021) FedSpeed (Sun et al., 2023), FedNTD (Lee et al., 2022), FedSAM (Caldarola et al., 2022) and FedDisco (Ye et al., 2023) and consistently observe performance improvement.

In summary, the key contributions of this work are:

- We introduced a novel computationally efficient regularization method ASD in the context of Federated Learning that alleviates the client drift problem by adaptively weighting the regularization loss for each sample based on the global model's prediction entropy and the label distribution of client data.

- We demonstrate the efficiency of our method by extensive experiments on datasets such as CIFAR-10, CIFAR-100, and Tiny-ImageNet datasets by combining our proposed ASD regularizer with the popular FL methods and improving their performance.

- We present a theoretical analysis of the client-drift and show that our regularizer minimizes the client-drift. We also empirically show that ASD promotes better generalization by converging to a flat minimum.

## 2 Related Work

### 2.1 Federated Learning (FL)

In recent times, addressing heterogeneity in Federated Learning has become an active area of research, and the field is developing rapidly. For brevity, we discuss a few related works here. In FedAvg (McMahan et al., 2017), the two main challenges explored are reducing communication costs (Yadav & Yadav, 2016) and ensuring privacy by avoiding having to share the data. There are some studies based on gradient inversion (Geiping et al., 2020) raising privacy concerns owing to gradient sharing while some studies have proposed in defense of sharing the gradients (Kairouz et al., 2021; Huang et al., 2021). FedAvg is the generalization of local SGD (Stich, 2018) by increasing the number of local updates, significantly reducing communication costs for an iid setting, but does not give similar improvements for non-iid data. Several works perform

an SGD-type analysis that involves the full device participation, and this breaks the important constraint in FL setup of partial device participation. Some of these attempt to compress the models to reduce the communication cost (Mishchenko et al., 2019). A few works include regularization methods on the client side (Zhu et al., 2021), and one-shot methods where clients send the condensed data and the server trains on the condensed data (Zhou et al., 2020). In (Hsu et al., 2020), an adaptive weighting scheme is considered on task-specific loss to minimize the learning from samples whose representation is negligible. Flatness-based methods based on SAM called as FedSAM is introduced in (Qu et al., 2022; Caldarola et al., 2022).

## 2.2 Client-Drift in FL

Due to data heterogeneity, federated training suffers from client drift. To address this, momentum-based server aggregation was proposed in (Wang et al.; Hsu et al., 2019), which was later extended to handle any client and server updates in (Reddi et al.) FedProx (Li et al., 2020) introduced a proximal term to penalize deviations of client weights from the globally initialized model. SCAFFOLD (Karimireddy et al., 2020) tackled the issue as one of objective inconsistency, introducing a gradient correction term as a regularizer. Subsequently, FedDyn (Acar et al., 2021) enhanced this with a dynamic regularization term. In (Kim et al., 2024), a proximal term was introduced in the client's optimization based on the accelerated global model, and momentum was applied on the server to track its updates.

## 2.3 Federated Learning Using Knowledge Distillation

Knowledge Distillation (KD) introduced by (Hinton et al., 2015) is a technique to transfer the knowledge from a pre-trained teacher model to the student model by matching the predicted probabilities. Self-distillation was introduced in (Zhang et al., 2019) where the student distills from the same model to the sub-networks of the model. The teacher model predictions are updated every batch. In our method, distillation happens with the full network, and the teacher's predictions are updated after every communication round. Adaptive distillation was used in (Tang et al., 2019). The server-side KD methods such as FedGen (Seo et al., 2022) use KD to train the generator at the server and the generator is broadcasted to the clients in the subsequent round. The clients use the generator to generate the data to provide the inductive bias. This method incurs extra communication of generator parameters along the model and training of the generator in general is difficult. In FedDF (Lin et al., 2020) KD is used at the server that relies on the external data. The KD is performed on an ensemble of client models, especially client models acts as a separate teacher model and then the knowledge is distilled into a single student model (global model). In FedNTD (Lee et al., 2022) the non-true class logits are used for distillation. This method gives uniform weights to all the samples. In FedCAD (He et al., 2022b) and FedSSD (He et al., 2022a), the client-drift problem is posed as a forgetting problem, and a weighting scheme has been proposed. Importantly, the computation of adaptive weights of the client samples is done with the help of the server with the assumption that the server has access to auxiliary data. One shortcoming of this method is the assumption of the availability of auxiliary data on the server, which is impractical. In (Zhang et al., 2022) logits were calibrated based on the label distribution. This is totally different from our approach as we are adjusting the weights of the distillation loss. Unlike all of these approaches, we propose a novel ASD strategy that aims to mitigate the challenge of client drift due to non-iid data without relying on the server and access to any form of auxiliary data to compute the adaptive weights.

# 3 Method

We first describe the traditional federated optimization problem, then explain the proposed method of adaptive self-distillation (ASD) in section 3.2. We provide the theoretical and empirical analysis in the sections 3.3 and 3.4 respectively.

## 3.1 Problem Setup

We assume there is a single server/cloud and $m$ clients/edge devices. We further assume that client $k$ has its own training dataset $\mathcal{D}_k$ with $n_k$ training samples drawn iid from the data distribution $\mathbb{P}_k(x, y)$. The data distributions $\{\mathbb{P}_k(x, y)\}_{k=1}^{K}$ across the clients are assumed to be non-iid. In this setup, we perform the

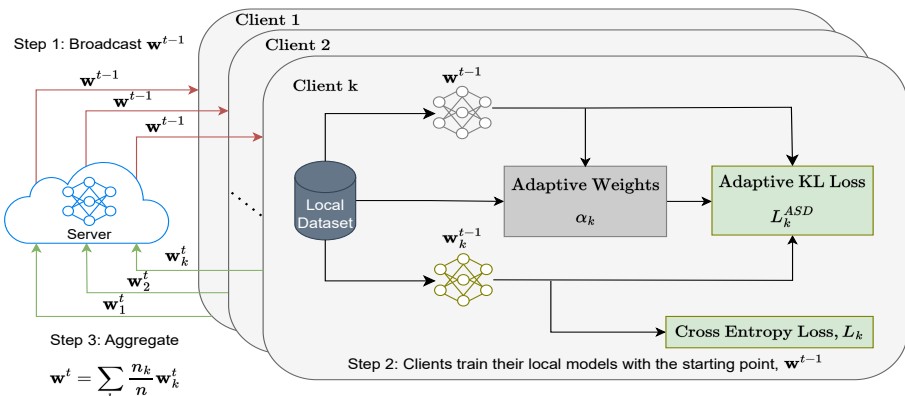

Figure 2: Federated Learning with Adaptive Self-Distillation: The figure describes the overview of the proposed approach based on Adaptive distillation. In **Step 1**. The server broadcasts the model parameters, In **Step 2**. clients train their models by minimizing both the cross entropy loss and predicted probability distribution over the classes between the global model and the client model by minimizing the KL divergence, the importance of each sample in the batch is decided by the proposed adaptive scheme as a function of label distribution and the KL term. The server model is fixed while training the client. In **Step 3**. The server aggregates the client models based on FedAvg aggregation. The process repeats till convergence.

following optimization. (Acar et al., 2021; McMahan et al., 2017)

$$\underset{w \in R^d}{arg\,min} \left( f(\mathbf{w}) \triangleq \frac{1}{K} \sum_{k \in [K]} f_k(\mathbf{w}) \right) \tag{1}$$

where $f_k(\mathbf{w})$ is the client specific objective function and $\mathbf{w}$ denotes model parameters. The overall FL framework is described in detail in figure 2.

### 3.2 Adaptive Self-Distillation (ASD) in FL

We now describe the proposed method where each client $k$ minimizes the $f_k(\mathbf{w})$ as defined below Eq. (2).

$$f_k(\mathbf{w}) \triangleq L_k(\mathbf{w}) + \lambda L_k^{ASD}(\mathbf{w}) \tag{2}$$

$L_k(\mathbf{w})$ is given below.

$$L_k(\mathbf{w}) = \underset{x,y \in P_k(x,y)}{E}[l_k(\mathbf{w}; (x, y))] \tag{3}$$

Here, $l_k$ is cross-entropy loss. The expectation is computed over training samples drawn from $\mathbb{P}_k(x, y)$ of a client $k$. This is approximated as the empirical average of the losses corresponding to samples from the Dataset $\mathcal{D}_k$. $L_k^{ASD}(\mathbf{w})$ in Eq. 2 denotes our proposed Adaptive Self-Distillation loss (ASD) term which considers label imbalance and quantifies how easily the predictions of the local model can drift from the global model. ASD loss is designed so that client models learn from the local data and at the same time not drift too much from the global model. We define (ASD) Loss as follows.

$$L_k^{ASD}(\mathbf{w}) \triangleq \mathbb{E}[\alpha_k(x, y)\mathcal{D}_{\text{KL}}(q_g(x, \mathbf{w}^t)||q_k(x, \mathbf{w}))] \tag{4}$$

In the above Eq. 4 $\mathbf{w}^t$ represents the global model parameters at FL round $t$ and $\mathbf{w}$ represents the trainable model parameters of client $k$, initialized with $\mathbf{w}^t$ at round $t$. $\alpha_k(x, y)$ denotes the weight for the sample $x$ with label ground truth label $y$. For simplicity, we denote the global model softmax predictions $q_g(x, \mathbf{w}^t)$ as $q_g(x)$ and client model softmax predictions $q_k(x, \mathbf{w})$ as $q_k(x)$. $\mathcal{D}_{\text{KL}}$ is the KL divergence. The Eq. 4 can be approximated by the following equation for a mini-batch.

$$L_k^{ASD}(\mathbf{w}) = \frac{1}{B} \sum_{i \in [B]} \alpha_k(x^i, y^i)\mathcal{D}_{\text{KL}}(q_g(x^i)||q_k(x^i)) \tag{5}$$

where $B$ is the batch size, $(x^i, y^i) \in \mathcal{D}_k$, $q_g$ and $q_k$ are softmax probabilities on the temperature ($\tau$) scaled logits of the global model and client model $k$ respectively. For a class $c$ below Eq. 6 and Eq. 7 holds.

$$q_g^c(x^i) = \frac{exp\left(z_g^c(x^i)/\tau\right)}{\sum_{m \in C} \exp\left(z_g^m(x^i)/\tau\right)} \tag{6}$$

$$q_k^c(x^i) = \frac{exp(z_k^c(x^i)/\tau)}{\sum_{m \in C} \exp\left(z_k^m(x^i)/\tau\right)} \tag{7}$$

where $z_g(x^i)$, $z_k(x^i)$ are the logits predicted on the input $x^i$ by the global model and client model $k$ respectively. The index $i$ denotes the $i^{th}$ sample of the batch. The $\mathcal{D}_{\mathrm{KL}}(q_g(x^i)||q_k(x^i))$ is given in Eq. (8).

$$\mathcal{D}_{\mathrm{KL}}(q_g(x^i)||q_k(x^i)) = \sum_{c=1}^{C} q_g^c(x^i) log(q_g^c(x^i)/q_k^c(x^i)) \tag{8}$$

where $C$ is the number of classes. We use the simplified notation $\alpha_k^i$ for distillation weights $\alpha_k(x^i, y^i)$ and it is given in below Eq.9.

$$\alpha_k^i = \frac{\hat{\alpha_k}^i}{\sum_{i \in B} \hat{\alpha_k}^i} \tag{9}$$

and $\hat{\alpha_k}^i$ is defined as below Eq. 10

$$\hat{\alpha_k}^i \triangleq \frac{exp(-\mathcal{H}(x^i))}{p_k^{y^i}} \tag{10}$$

where $\mathcal{H}(x^i)$ is the entropy of the global model predictions and is given by (11).

$$\mathcal{H}(x^i) = \sum_{c=1}^{C} -q_g^c(x^i) log(q_g^c(x^i)) \tag{11}$$

$\mathcal{D}_{\mathrm{KL}}$ in Eq. 8 captures how close the local model's predictions are to the global model for any given sample $x^i$. Our weighting scheme in Eq. 11 decides how much to learn from the global model for that sample based on the entropy $\mathcal{H}(x^i)$ of the server model predictions and the label distribution of the client data ($p_k^{y^i}$). $\mathcal{H}(x^i)$ captures the confidence of global model predictions, higher value implies the server predictions are noisy so we tend to reduce the weight, i.e, we give less importance to the global model if its entropy is high. The $p_k^{y^i}$ is the probability that the sample belong to a particular class. We give more weight to the sample if it belongs to the minority class. This promotes learning from the local data for the classes where the representation is sufficient enough and for the minority classes we encourage them to stay closer to the global model. In summary, the choice of alpha is designed to ensure that, when the global model encounters samples with high prediction entropy, we decrease the weighting on the regularization loss. Conversely, for samples with a low probability of occurrence, we prioritize learning from the global model. This adaptive approach enables local models to effectively learn from the cross-entropy loss for more frequent labels while leveraging the global model's guidance for less frequent labels. In Table 3, we highlight the importance of adaptive weights, where we clearly show that ASD with adaptive weights consistently improves performance when combined with off-the-shelf FL methods. We approximate the label distribution $p_k^{y^i}$ with the empirical label distribution, it is computed as Eq.12.

$$p_k^{y^i=c} = \frac{\sum_{i \in |\mathcal{D}_k|} \mathbb{I}_{y^i=c}}{|\mathcal{D}_k|} \tag{12}$$

where $\mathbb{I}_{y^i=c}$ denotes the indicator function and its value is 1 if the label of the $i^{th}$ training sample belongs to class $c$ else it is 0. To simplify notation, we use $p_k^c$ for $p_k^{y^i=c}$ as it depends only on the class $c$ for a client $k$. Finally we use Eq.12 and Eq. 11 to compute the $L_k^{ASD}(\mathbf{w})$ defined in Eq.5. The choice of KL divergence in the Eq. 8 is motivated by the seminal work of Hinton et.al., which aims to match the temperature-raised softmax values between the pre-trained teacher model and student model for effective knowledge transfer. We also analyzed the other statistical divergences such as reverse KL and Jenson-shannon divergence and empirically found that KL divergence is better. More details are presented in the Sec. A.13 of the Appendix.

### 3.3 Theoretical Analysis of Gradient Dissimilarity

In this section, we perform the theoretical analysis of the client drift. We now introduce the Gradient dissimilarity $G_d$ based on the works of (Li et al., 2020; Lee et al., 2022) as a way to measure the extent of client-drift as below.

$$G_d(\mathbf{w}, \lambda) = \frac{\frac{1}{K} \sum_k \|\nabla f_k(\mathbf{w})\|^2}{\|\nabla f(\mathbf{w})\|^2} \tag{13}$$

$G_d(\mathbf{w}, \lambda)$ is function of both the $\mathbf{w}$ and $\lambda$. For convenience, we simply write $G_d$ and mention arguments explicitly when required. $f_k(\mathbf{w})$ in the above Eq. 13 is same as Eq. 2.

With this, we now establish a series of propositions to show that ASD regularization reduces the Gradient dissimilarity, which as a result, leads to lower client drift.

**Proposition 3.1.** $\inf_{\mathbf{w} \in \mathbb{R}^d} G_d(\mathbf{w}, \lambda)$ *is* 1, $\forall \ \lambda$

The above proposition implies that if all the client's gradients are progressing in the same direction, which means there is no drift $G_d = 1$. The result follows from Jensen's inequality. The lower value of $G_d$ is desirable and ideally 1. To analyze the $G_d$, we need $\nabla f_k(\mathbf{w})$ which is given in the below proposition.

**Proposition 3.2.** *When the class conditional distribution across the clients is identical, i.e., $\mathbb{P}_k(x \mid y) = \mathbb{P}(x \mid y)$ then $\nabla f_k(\mathbf{w}) = \sum_c p_k^c(\mathbf{g}_c + \lambda \gamma_k^c \tilde{\mathbf{g}}_c)$, where $\mathbf{g}_c = \nabla \mathbb{E}[l(\mathbf{w}; x, y) \mid y = c]$, $\tilde{\mathbf{g}}_c = \nabla \mathbb{E}[\exp(-\mathcal{H}(x)) \mathcal{D}_{KL}(q_g(x) \| q_k(x)) \mid y = c]$ and $\gamma_k^c = \frac{1}{p_k^c}$.*

The result follows from the tower property of expectation and the assumption that class conditional distribution is the same for all the clients. From the above proposition, we can see that the gradients $\nabla f_k(\mathbf{w})$ only differ due to $p_k^c$ which captures the data heterogeneity due to label imbalance. The proof is given in Sec. A.15 of the appendix.

**Assumption 3.3.** *Class-wise gradients are weakly correlated and similar magnitude* $\mathbf{g}_c^\intercal \mathbf{g}_c \ll \mathbf{g}_c^\intercal \mathbf{g}_m$, $\tilde{\mathbf{g}}_c^\intercal \tilde{\mathbf{g}}_c \ll \tilde{\mathbf{g}}_c^\intercal \tilde{\mathbf{g}}_m$ *and for* $c \neq m$

The assumption on weakly correlated class-wise gradients intuitively implies that gradients of loss for a specific class cannot give any significant information on the gradients of the other class.

**Proposition 3.4.** *When the class-conditional distribution across the clients is the same, and the Assumption 3.3 holds then $\exists$ a range of values for $\lambda$ such that whenever $\lambda \geq \lambda_c$ we have $\frac{dG_d}{d\lambda} < 0$ and $G_d(\mathbf{w}, \lambda) < G_d(\mathbf{w}, 0)$.*

The proposition implies that there is a value of $\lambda \geq \lambda_c$ such that the derivative of $G_d$ w.r.t $\lambda$ is negative. The proof is given in Sec. A.15 of the appendix. This indicates that by appropriately selecting the value of $\lambda$ we can make the $G_d$ lower which in turn reduces the client drift. One of the key assumptions on the heterogeneity is the existence of the below quantity.

$$B^2(\lambda) := \sup_{\mathbf{w} \in \mathbb{R}^d} G_d(\mathbf{w}, \lambda) \tag{14}$$

which leads to the following assumption

**Assumption 3.5.** $\frac{1}{K} \sum_k \|\nabla f_k(\mathbf{w})\|^2 \leq B^2(\lambda) \|\nabla f(\mathbf{w})\|^2$

This is the bounded gradient dissimilarity assumption used in (Li et al., 2020). In the following proposition, we show the existence of $\lambda$ such that $B^2(\lambda) < B^2(0)$, which means that with regularizer we can tightly bound the gradient dissimilarity compared to the case without the regularizer i.e., ($\lambda = 0$).

**Proposition 3.6.** *Suppose the functions $f_k$ satisfy Assumption 3.5 above then we have $B^2(\lambda) < B^2(0)$.*

*Proof.* From 14 we have

$$B^2(\lambda) = \sup_{\mathbf{w} \in \mathbb{R}^d} G_d(\mathbf{w}, \lambda) \tag{15}$$

For a fixed $\lambda$ as per proposition 3.4 we have the following.

$$\sup_{\mathbf{w}\in\mathbb{R}^d} G_d(\mathbf{w},\lambda) < \sup_{\mathbf{w}\in\mathbb{R}^d} G_d(\mathbf{w},0) \tag{16}$$

The above inequality 16 is true as proposition 3.4 guarantees that the value of $G_d(\mathbf{w},\lambda) < G_d(\mathbf{w},0)$ for all $\mathbf{w}$ when $\lambda \geq \lambda_c$. If inequality 16 is not true, one can find a $\mathbf{w}$ that contradicts the proposition 3.4 which is impossible. This means for some value of $\lambda \geq \lambda_c$ we have $B^2(\lambda) < B^2(0)$ from Eq. 14 and Eq. 16. $\qquad\square$

The key takeaway from the analysis is that by introducing the regularizer we can tightly bound the heterogeneity when compared to the case without the regularizer. *Based on the works (Karimireddy et al., 2020; Li et al., 2020) we explain that lower $B^2(\lambda)$ implies better convergence, which is also supported by empirical evidence. These details are provided in the Sec. A.16 of the Appendix.*

### 3.4 Discussison on the Generalization of ASD

Table 1: The table shows the impact of ASD on the algorithms on CIFAR-100 Dataset using the non-iid partition of $\delta = 0.3$. We consistently see that the top eigenvalue and the trace of the Hessian of the loss of the global model decrease and the accuracy improves when ASD is used. This suggests that by using ASD we can make global model reach a flat minimum towards better generalization.

| Algorithm | Top Eigenvalue ↓ | Trace ↓ | Accuracy ↑ |
|---|---|---|---|
| FedAvg | 53.6 | 8516 | 38.67 |
| FedAvg + ASD | **12.3** | **2269** | **42.77** |
| FedDyn | 49.4 | 6675 | 47.56 |
| FedDyn + ASD | **14.2** | **2241** | **49.03** |
| FedSpeed | 51.9 | 6937 | 47.39 |
| FedSpeed + ASD | **14.6** | **2063** | **49.16** |

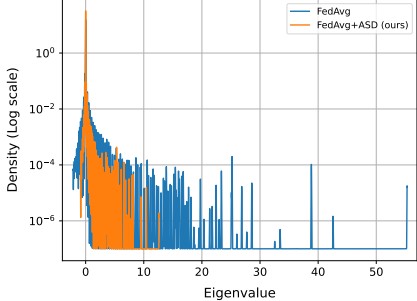

Figure 3: Eigen spectrum with and without the ASD regularizer. It is evident that ASD regularizer not only minimizes the top eigenvalue but most of the eigenvalues and attains the flatness.

The key reason for better generalization is the adaptive self-distillation loss. It has been shown in (Mobahi et al., 2020) that self-distillation improves the generalization in centralized settings. It's been empirically shown in (Zhang et al., 2019) that self-distillation helps the model to converge to flat-minimum. Generally, converging to flat minima is indicative of improved generalization, a concept explored in prior studies such as (Keskar et al., 2017) and (Yao et al., 2020). The top eigenvalue and the trace of the Hessian computed from the training loss are typical measures of 'flatness' of the minimum to which the training converges, i.e., lower values of these measures indicate the presence of a flat minimum. To gain a deeper understanding of this phenomenon in a federated learning setting, we analyzed the top eigenvalue and the trace of the Hessian of the cross-entropy loss for global models obtained with and without the ASD regularizer. The following argument establishes that if the client models converge to flat minima, it would also ensure convergence of the resultant global model to a flat minimum. We assume the Hessians of the functions $f_k$ ($k^{th}$ client's local objective),

$f$ (resultant global objective) exist and are continuous almost everywhere. Since $f = \frac{1}{K}\sum_{i=1}^{K} f_i$, we have $\mathbf{H}(f) = \frac{1}{K}\sum_{i=1}^{K} \mathbf{H}(f_i)$ ($\mathbf{H}(g)$ denotes the Hessian of function $g$). This implies $\mu_1(\mathbf{H}(f)) \leq \frac{1}{K}\sum_{i=1}^{K} \mu_1(\mathbf{H}(f_i))$ ($\mu_1(\mathbf{A})$ denotes top eigenvalue of matrix $\mathbf{A}$). Thus when the local models converge to a flat minimum, it will ensure the convergence of the global model to a flat minimum. Following the method of (Yao et al., 2020), we computed the top eigenvalue and trace of the Hessian. In Table 1, we observe that FedAvg+ASD attains lower values for the top eigenvalue and trace compared to FedAvg, suggesting convergence to flat minimum. The Eigen density plot in the figure 3 also confirms the same. We use the CIFAR-100 dataset with non-iid data partitioning of $\delta = 0.3$ (refer to Sec. 4). In Table 1 we have presented our analysis when ASD is combined with FedAvg, FedDyn and FedSpeed. The results for other algorithms are presented in Sec. A.8 of appendix. A similar concept has been explored in FedSAM (Qu et al., 2022; Caldarola et al., 2022); the issue with SAM-based methods is they require an extra forward and backward pass, which doubles the computational cost on the resource constrained edge devices. However, our method can be applied to SAM-based methods and further improve its performance. ASD consistently attains the flatness with the other FL algorithms and enhances their generalization.

## 4 Experiments

We perform the experiments on CIFAR-10, CIFAR-100 (Krizhevsky & Hinton, 2009), Tiny-ImageNet (Le & Yang, 2015) datasets with different degrees of heterogeneity in the balanced settings (i.e., the same number of samples per client but the class label distribution of each varies). We set the total number of clients to 100 in all our experiments. We set the client participation rate to 0.1, i.e., 10 percent of clients are sampled on an average per communication round, similar to the protocol followed in (Acar et al., 2021). We build our experiments using publicly available codebase by (Acar et al., 2021). For generating non-iid data, Dirichlet distribution is used. To simulate the effect of label imbalance, for every client we sample the 'probability distribution' over the classes from the aforementioned Dirichlet distribution $p_k^{dir} = Dir(\delta, C)$. Every sample of $p_k^{Dir}$ is a vector of length $C$ and all the elements of this vector are non-negative and sum to 1. This vector represents the label distribution for the client. The parameter $\delta$ known as the 'concentration parameter', captures the degree of label heterogeneity. Lower values of $\delta$ capture high heterogeneity and as the value of $\delta$ increases, the label distribution becomes more uniform. Another parameter of Dirichlet distribution (i.e., $C$), its value can be interpreted from the training dataset ($C = 100$ for CIFAR-100). For notational convenience, we omit $C$ from $Dir(\delta, C)$ by simply re-writing as $Dir(\delta)$. By configuring the concentration parameter $\delta$ to 0.6 and 0.3, we sample the data using the Dirichlet distribution across the labels for each client from moderate to high heterogeneity by controlling $\delta$. This is in line with the approach followed in (Acar et al., 2021) and (Yurochkin et al., 2019).

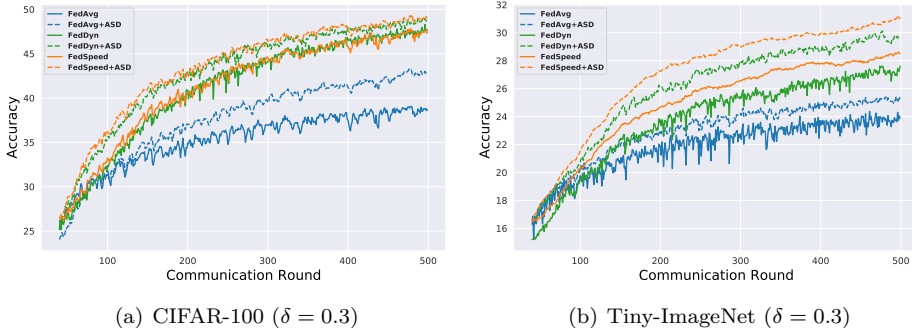

(a) CIFAR-100 ($\delta = 0.3$)  (b) Tiny-ImageNet ($\delta = 0.3$)

Figure 4: Test Accuracy vs Communication rounds: Comparison of algorithms with $\delta = 0.3$ partitions on CIFAR-100 and Tiny-ImageNet datasets. All the algorithms augmented with proposed regularization (ASD) outperform compared to their original form. FedSpeed+ASD outperforms all the other algorithms.

Table 2: Comparison of Accuracy(%): We show the accuracy attained by the algorithms across the datasets (CIFAR-100/Tiny-ImageNet) at the end of 500 communication rounds. It can be seen that by combining the proposed approach the performance of all the algorithms can be significantly improved.

| Algorithm | CIFAR-100 | | | TinyImageNet | | |
|---|---|---|---|---|---|---|
| | $\delta = 0.3$ | $\delta = 0.6$ | IID | $\delta = 0.3$ | $\delta = 0.6$ | IID |
| FedAvg (McMahan et al., 2017) | $38.67_{\pm 0.66}$ | $38.53_{\pm 0.32}$ | $37.68_{\pm 0.41}$ | $23.89_{\pm 0.84}$ | $23.95_{\pm 0.72}$ | $23.48_{\pm 0.61}$ |
| FedAvg+ASD (**Ours**) | $\mathbf{42.77}_{\pm 0.22}$ | $\mathbf{42.54}_{\pm 0.51}$ | $\mathbf{43.00}_{\pm 0.60}$ | $\mathbf{25.31}_{\pm 0.25}$ | $\mathbf{26.38}_{\pm 0.21}$ | $\mathbf{26.67}_{\pm 0.10}$ |
| FedProx (Li et al., 2020) | $37.79_{\pm 0.97}$ | $37.92_{\pm 0.55}$ | $37.94_{\pm 0.22}$ | $24.61_{\pm 1.24}$ | $23.57_{\pm 0.44}$ | $23.27_{\pm 0.11}$ |
| FedProx+ASD (**Ours**) | $\mathbf{41.31}_{\pm 0.90}$ | $\mathbf{41.67}_{\pm 0.12}$ | $\mathbf{42.30}_{\pm 0.37}$ | $\mathbf{25.49}_{\pm 0.45}$ | $\mathbf{25.62}_{\pm 0.05}$ | $\mathbf{25.58}_{\pm 0.18}$ |
| FedNTD (Lee et al., 2022) | $40.40_{\pm 1.52}$ | $40.50_{\pm 0.54}$ | $41.23_{\pm 0.44}$ | $23.71_{\pm 0.65}$ | $23.28_{\pm 0.29}$ | $22.95_{\pm 0.22}$ |
| FedNTD+ASD (**Ours**) | $\mathbf{43.01}_{\pm 0.34}$ | $\mathbf{43.61}_{\pm 0.33}$ | $\mathbf{43.25}_{\pm 0.41}$ | $\mathbf{27.34}_{\pm 0.73}$ | $\mathbf{27.39}_{\pm 0.39}$ | $\mathbf{27.41}_{\pm 0.11}$ |
| FedDyn (Acar et al., 2021) | $47.56_{\pm 0.41}$ | $48.60_{\pm 0.09}$ | $48.87_{\pm 0.51}$ | $27.62_{\pm 0.21}$ | $28.58_{\pm 0.61}$ | $28.37_{\pm 0.20}$ |
| FedDyn+ASD (**Ours**) | $\mathbf{49.03}_{\pm 0.24}$ | $\mathbf{50.23}_{\pm 0.25}$ | $\mathbf{51.44}_{\pm 0.48}$ | $\mathbf{29.94}_{\pm 0.67}$ | $\mathbf{30.05}_{\pm 0.24}$ | $\mathbf{30.76}_{\pm 0.44}$ |
| FedSAM (Caldarola et al., 2022) | $40.89_{\pm 0.30}$ | $41.41_{\pm 0.34}$ | $40.81_{\pm 0.26}$ | $24.72_{\pm 0.64}$ | $25.42_{\pm 0.49}$ | $23.50_{\pm 0.94}$ |
| FedSAM+ASD (**Ours**) | $\mathbf{43.99}_{\pm 0.14}$ | $\mathbf{44.54}_{\pm 0.30}$ | $\mathbf{44.77}_{\pm 0.11}$ | $\mathbf{26.26}_{\pm 0.47}$ | $\mathbf{26.80}_{\pm 0.17}$ | $\mathbf{25.37}_{\pm 0.26}$ |
| FedDisco (Ye et al., 2023) | $38.97_{\pm 1.38}$ | $38.87_{\pm 1.37}$ | $37.85_{\pm 0.57}$ | $24.35_{\pm 0.42}$ | $24.03_{\pm 0.78}$ | $23.49_{\pm 0.31}$ |
| FedDisco+ASD (**Ours**) | $\mathbf{41.55}_{\pm 1.06}$ | $\mathbf{41.94}_{\pm 0.30}$ | $\mathbf{43.09}_{\pm 0.47}$ | $\mathbf{25.43}_{\pm 0.46}$ | $\mathbf{26.03}_{\pm 0.26}$ | $\mathbf{26.56}_{\pm 0.9}$ |
| FedSpeed (Sun et al., 2023) | $47.39_{\pm 0.82}$ | $48.27_{\pm 0.13}$ | $49.01_{\pm 0.46}$ | $28.60_{\pm 0.15}$ | $29.33_{\pm 0.3}$ | $29.62_{\pm 0.31}$ |
| FedSpeed+ASD (**Ours**) | $\mathbf{49.16}_{\pm 0.40}$ | $\mathbf{49.76}_{\pm 0.27}$ | $\mathbf{51.99}_{\pm 0.32}$ | $\mathbf{30.97}_{\pm 0.25}$ | $\mathbf{30.05}_{\pm 0.24}$ | $\mathbf{32.68}_{\pm 0.53}$ |

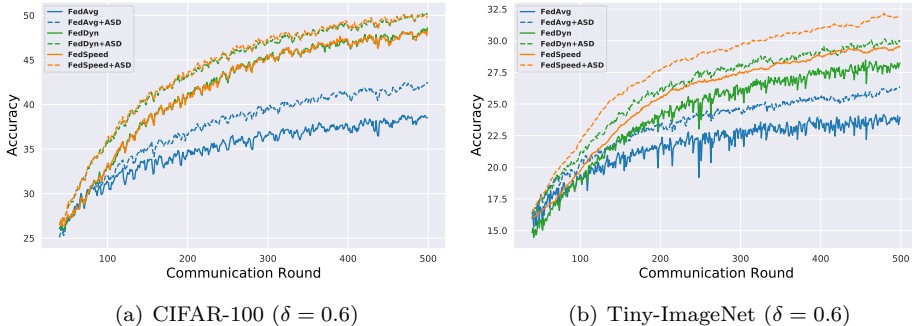

(a) CIFAR-100 ($\delta = 0.6$)  (b) Tiny-ImageNet ($\delta = 0.6$)

Figure 5: Test Accuracy vs Communication rounds: Comparison of algorithms with $\delta = 0.6$ data partitions on CIFAR-100 and Tiny-ImageNet dataset. All the algorithms augmented with proposed regularization (ASD) outperform compared to their original form. FedSpeed+ASD outperforms all the other algorithms.

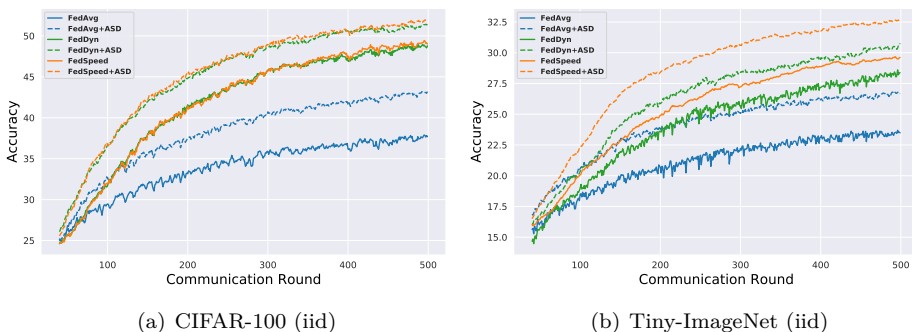

(a) CIFAR-100 (iid)  (b) Tiny-ImageNet (iid)

Figure 6: Test Accuracy vs Communication rounds: Comparison of algorithms with iid data partitions on CIFAR-100 and Tiny-ImageNet datasets. All the algorithms augmented with proposed regularization (ASD) outperform compared to their original form. FedSpeed+ASD outperforms all the other algorithms.

# 5 Results and Discussion

For evaluation, we report accuracy on the test dataset as our performance metric and the number of communication rounds required to attain the desired accuracy as a metric to quantify the communication cost. Specifically, we evaluate the global model on the test set and report its accuracy after every communication round. For comparison, we consider the popular methods for federated learning, such as FedAvg, FedProx, FedDyn, FedSpeed FedNTD, FedSAM and FedDisco. We augment each of these methods with our approach (ASD) and observe a significant boost in performance. For a fair comparison, we consider the same models used in Fedavg (McMahan et al., 2017), and FedDyn (Acar et al., 2021), for CIFAR-10 and CIFAR-100 classification tasks. The model architecture used for CIFAR-100 contains 2 convolution layers followed by 3 fully connected layers. For Tiny-ImageNet, we use 3 convolution followed by 3 fully connected layers. The detailed architectures are given in Sec A.2 of the appendix. Hyperparameters: SGD algorithm with a learning rate of 0.1 and decay the learning rate per round of 0.998 is used to train the client models. Temperature $\tau$ is set to 2.0. We only tune the hyper-parameter $\lambda$. More hyperparameter setting details and impact of $\lambda$, $\tau$ are provided in Sec. A.3 and A.6 of the appendix, respectively. The impact of client participation rate and the number of clients on ASD are shown in Sec. A.7 of the appendix. Implementation of ASD with other FL methods is discussed in Sec. A.12 of appendix. We compare the convergence of different schemes for 500 communication rounds. Following the testing protocol of (Acar et al., 2021), we average across all the client models and compute the test accuracy on the averaged model, which is reported in our results. In all the tables, we report the test accuracy of the global model in % at the end of 500 communication rounds. All the experiments in the tables are performed over three different initializations, mean and standard deviations of accuracy over the three experiments are reported. We also demonstrate the efficacy of our proposed method with deeper architectures such as ResNet-20 and Vision Transformer (ViT) models in Sec A.4 and Sec A.5 of the appendix respectively.

## 5.1 Performance of ASD on CIFAR-10/100 and Tiny-Imagenet

In Table 2, we report the performance of CIFAR-100 and Tiny-ImageNet datasets with various algorithms for non-iid ($Dir(\delta = 0.3)$ and $Dir(\delta = 0.6)$) as well as the iid settings. Each experiment is performed over three different initializations, and the mean and standard deviation of the accuracy are reported. On CIFAR-100, we observe that our proposed ASD applied on FedDyn improves its performance by $\approx 1.45\%$ for $Dir(\delta = 0.3)$ and $\approx 1.6\%$ $Dir(\delta = 0.6)$. Similarly, for Tiny-ImageNet we observe that FedDyn+ASD improves FedDyn by $\approx 2.4\%$ for $Dir(\delta = 0.3)$ and by $\approx 1.4\%$ for $Dir(\delta = 0.6)$. The test accuracy vs communication rounds plot on CIFAR-100 and Tiny-ImageNet datasets is shown in Figures 4, 5 6 across non-iid and iid partitions. We can see that adding ASD gives consistent improvement across the rounds[1]. We obtain significant improvements for FedAvg+ASD against FedAvg, FedProx+ASD against FedProx, FedSpeed+ASD against FedSpeed, etc. In Sec A.9 of the appendix we present the CIFAR-10 results where we observe that adding ASD consistently gives an improvement of $\approx 0.4\% - 1.99\%$ improvement across the algorithms.

## 5.2 Comparison with adaptive vs uniform weights

Table 3: Comparison with adaptive weights vs uniform weights on CIFAR-100 dataset with Dirichlet $\delta = 0.3$

| Algorithm | Distillation with Uniform weights | Distillation with Adaptive weights |
|---|---|---|
| FedAvg+ASD | 41.75 $_{\pm 0.12}$ | **42.77** $_{\pm 0.22}$ |
| FedNTD+ASD | 40.40 $_{\pm 1.52}$ | **43.01** $_{\pm 0.34}$ |
| FedDisco+ASD | 40.21 $_{\pm 0.57}$ | **41.55** $_{\pm 1.06}$ |
| FedDyn+ASD | 47.90 $_{\pm 0.35}$ | **49.03** $_{\pm 0.24}$ |

We analyze the impact of the proposed adaptive weighting scheme. We compare by making all the $\hat{\alpha_k}^i$ in Eq 10 to 1 i.e, by giving equal weights to all the samples in the mini-batch. We can see from Table 3 that the

---

[1]In the figures we only compare FedAvg, FedDyn and FedSpeed for better readability. For others, please refer to Sec A.10 of the appendix

proposed adaptive weighting scheme yields much better performance than assigning uniform weights, thus establishing the impact of proposed adaptive weights.

### 5.3 Performance with increased clients and lower client participation

Table 4: Experiments with 500 clients

| Method | Accuracy (in %) |
|---|---|
| FedAvg | 27.92 |
| FedAvg+ASD (ours) | **31.28** |
| FedProx | 28.09 |
| FedProx+ASD | **32.13** |
| FedNTD | 30.99 |
| FedNTD+ASD | **33.65** |
| FedDyn | 31.0 |
| FedDyn+ASD (ours) | **33.12** |
| FedSpeed | 34.08 |
| FedSpeed+ASD (ours) | **36.59** |

In this section, we analyze the impact of our ASD regularizer to mimic the cross-device setting. We increase the client participation to 500 clients and only 1% of the clients participate in every round. We consider the CIFAR-100 dataset and the non-iid data partition of $\delta = 0.3$. In the Table 4, we observe that ASD consistently improves the performance of the algorithms. The accuracies are reported after averaging over three different initializations at the end of 1000 communication rounds. Even in this challenging setting ASD consistently improves the performance of the FL algorithms.

## 6 Computation Cost

The major computation for the distillation scheme comes from the teacher forward pass, student forward pass, and the student backward pass (Xu et al., 2020). We assume $C_s$ as the total computational cost of server model forward pass and $C_k$ be the total computation cost of client model $k$ the forward pass per epoch. We do not need $C_s$ computations every epoch, we only need to compute once and store the values of $\mathcal{H}(x)$ while keeping the same backward computation. Specifically, for the computation of distillation regularizer we only need $E * C_k + C_s$ local computations compared to $E * C_k$ computations without regularizer. Here $E$ denotes the local epochs of the client. Since $C_s = C_k$, we have $(E + 1) * C_k$ local computations. Thus, our regularizer introduces minimal forward computation on the edge devices, which typically have low computation. In Sec. A.14 of appendix we discuss the computation vs accuracy of ASD.

## 7 Conclusion

In this work, we presented an efficient and effective method for addressing client data heterogeneity due to label imbalance in federated learning using our proposed Adaptive Self-Distillation (ASD), which does not require any auxiliary data and no extra communication cost. We also theoretically showed that ASD has lower client-drift leading to better convergence. Moreover, we performed analysis to show that ASD has better generalization by analyzing the top eigenvalue and trace of the Hessian of the global model's loss. The effectiveness of our approach is shown via extensive experiments across datasets such as CIFAR-10, CIFAR-100 and Tiny-ImageNet with different degrees of heterogeneity. Our proposed regularizer (ASD) can be integrated easily atop any of the FL frameworks. We evaluated this efficacy by showing improvement in the performance when combined with FedAvg, FedProx, FedDyn, FedSAM, FedDisco, FedNTD and FedSpeed. We have also shown that the computation required to implement ASD is simply an additional forward pass on the client-side training, i.e, all the gains we obtain with ASD requires minimal compute. Our research can inspire the designing of the computationally efficient regularizers that concurrently reduce client-drift and improve the generalization.

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

## A  Appendix

### A.1  Notations and Definitions

- $\mathcal{H}(x)$ denotes the entropy of the model under consideration for input $x$.
- $p_k^{y^i=c}$ denotes the probability that the client $k$ has the input $i$ belonging to class $c$.
- $\mathbf{H}(g)$ denotes the Hessian of the function $g$.
- $\mu_1(\mathbf{A})$ denotes the top eigenvalue of matrix $\mathbf{A}$.
- $\mathcal{D}_{KL}$ denotes the KL divergence.
- inf denotes the infimum and sup denotes the supremum.
- $\delta$ is used for denoting the heterogeneity generated based on Dirichlet distribution.
- $\lambda$ denotes the ASD regularizer strength.
- $G_d(\mathbf{w}, \lambda)$ denotes the gradient dissimilarity.
- $\mathbb{E}(.)$ denotes the expectation.
- $L_k(\mathbf{w})$ is the loss of client $k$ (cross-entropy loss).

- $L_k(\mathbf{w})^{ASD}$ is the Adaptive Self-Distillation loss for client $k$.

- $\mathcal{D}_k$ represents the dataset of client $k$.

- $\mathcal{P}_k(x, y)$ represents the data distribution of the client $k$.

## A.2    Model Architectures

In Table 5, the model architecture is shown. We use PyTorch style representation. For example conv layer(3,64,5) means 3 input channels, 64 output channels and the kernel size is 5. Maxpool(2,2) represents the kernel size of 2 and a stride of 2. FullyConnected(384,200) represents an input dimension of 384 and an output dimension of 200. The architecture for CIFAR-100 is exactly the same as used in (Acar et al., 2021).

Table 5: Models used for Tiny-ImageNet and CIFAR-100 datasets.

| CIFAR-10/100 Model | Tiny-ImageNet Model |
|---|---|
| | ConvLayer(3,64,3) |
| | GroupNorm(4,64) |
| | Relu |
| | MaxPool(2,2) |
| | ConvLayer(64,64,3) |
| | GroupNorm(4,64) |
| ConvLayer(3,64,5) | Relu |
| Relu | MaxPool(2,2) |
| MaxPool(2,2) | ConvLayer(64,64,3) |
| ConvLayer(64,64,5) | GroupNorm(4,64) |
| Relu | Relu |
| MaxPool(2,2) | MaxPool(2,2) |
| Flatten | Flatten |
| FullyConnected(1600,384) | FullyConnected(4096,512) |
| Relu | Relu |
| FullyConnected(384,192) | FullyConnected(512,384) |
| Relu | Relu |
| FullyConnected(192,100) | FullyConnected(384,200) |

## A.3    Hyper-Parameter Settings

The value of $\lambda$ is specified in units of batch-size $B$. We chose $\lambda$ from $\{10, 20, 30\}$. We set $\lambda = 20$ for all the Tiny-ImageNet experiments. For CIFAR-10/100 we chose $\lambda$ to be 10 and 30 respectively. The batch-size ($B$) of 50 and learning rate of 0.1 with decay of 0.998 is employed for all the experiments unless specified. All the experiments are carried out with 100 clients and with 10% client participation.

## A.4    Experiments with Deeper Models (CNN's)

In this section, we perform experiments with the deep models such as ResNet-20 on CIFAR-100 dataset with Dirichlet $\delta = 0.3$. For this experiment we have used 300 communication rounds, the number of clients as 30, and the client participation rate is set to 20%. In the table 6, we report the numbers averaged over 3 different trials. We observe that the addition of our proposed regularizer ASD atop mutiple popular FL methods leads to consistent improvements, thereby further justifying the efficacy of our proposed method.

Table 6: Experiments on ResNet-20

| Method | Accuracy (in %) |
|---|---|
| FedAvg | 46.35 |
| FedAvg+ASD (ours) | **47.90** |
| FedDyn | 53.60 |
| FedDyn+ASD (ours) | **55.15** |
| FedSpeed | 54.42 |
| FedSpeed+ASD (ours) | **55.82** |

### A.5   Experiments with Deeper Models (ViT)

We perform experiments with ViT architecture using the Tiny-ViT (Wu et al., 2022) as client models on ImageNet-100 dataset (Zang et al., 2022) with non-iid data partitioning of Dirichlet $\delta = 0.3$. The choice of Tiny-ViT is motivated by the fact that edge devices are traditionally computational resource-constrained and Tiny-ViT is designed for such applications. For this experiment, the number of clients is set to 200, and the client participation rate is set to 5%. We have used 300 communication rounds. In the Table 7, we report the numbers averaged over 3 different trials. We observe that the addition of our proposed regularizer ASD atop FedAvg and FedDyn leads to consistent improvements, thereby further justifying the efficacy of our proposed method on the deeper architectures.

Table 7: Experiments using Tiny-ViT on ImageNet-100 dataset with the non-iid partitioning of $\delta = 0.3$

| Method | Accuracy (in %) |
| --- | --- |
| FedAvg | 18.12 |
| FedAvg+ASD (ours) | **22.10** |
| FedDyn | 28.02 |
| FedDyn+ASD (ours) | **36.70** |

In Table 8, we performed an experiment on ViT-Small architecture on CIFAR-100. We observe that adding our ASD regularizer improves the baseline FedAvg by 1.4% and 1.7% for $\delta = 0.3$ and $\delta = 0.6$, respectively. In this setup we consider 100 clients with 10% participation and the accuracy is reported at the end of 300 rounds.

Table 8: Experiments using ViT-Small with CIFAR-100 with the non-iid data partitioning of $\delta = 0.3$ and $\delta = 0.6$.

| Method | Accuracy(%) | |
| --- | --- | --- |
| | $\delta = 0.3$ | $\delta = 0.6$ |
| FedAvg | 53.22 | 52.63 |
| FedAvg+ASD (Ours) | **54.67** | **54.34** |

### A.6   Impact on the choice of hyperparameters $\lambda$ and $\tau$

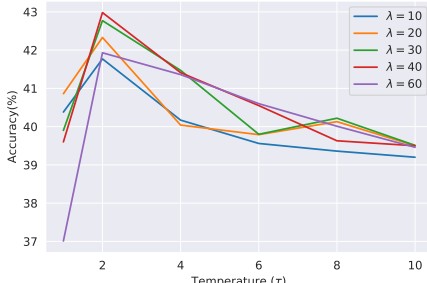

Figure 7: Impact of $\lambda$ and $\tau$ on CIFAR-100 dataset with non-iid partitioning of $\delta = 0.3$ with FedAvg+ASD.

We study the impact of changing the hyper-parameters $\lambda$ and $\tau$ on the CIFAR-100 dataset with the Dirichlet non-iid partition of $\delta = 0.3$. We report the accuracy at the end of 500 rounds. When using FedAvg+ASD algorithm. In Figure 7 we see that the accuracy of the model increases with $\lambda$ and then slightly drops after a critical point. This is expected as too less value of $\lambda$ is similar to FedAvg and very high value of $\lambda$ will ignore the local learning. It can also be seen that for all the values of $\lambda$ the Accuracy peaks at $\tau = 2$. In all of our experiments we set the temperature parameter $\tau$ set to 2.0.

### A.7 Impact of client participation / number of clients on ASD

We fix the client participation to 2% and vary the number of clients from 100 to 500. We perform this ablation using the CIFAR-100 dataset with a non-iid Dirichlet data partitioning of $\delta = 0.3$. We summarize our observations in the Tables 9 and 10 below. It can be seen that ASD improves the performance of the baselines FedAvg and FedDyn in all the settings. In particular, we would like to highlight the point here that despite increasing the number of clients, the total number of training data samples across all the clients remains constant (for CIFAR-100). Thus as the number of clients increases, the number of data samples per client decreases. This further aggravates the adverse impact of label heterogeneity across clients, and hence accuracy degrades in general. However, we are happy to observe and report that, even under such a challenging setup, our proposed adaptive self-distillation-based strategy consistently improves the accuracy when combined on top of the existing baseline algorithms.

Table 9: Impact of increasing the number of clients on the Accuracy (%) when the participation rate is fixed to 2% and **non-iid partitioning of** $\delta = 0.3$.

| Method | Number of Clients | | | | |
|---|---|---|---|---|---|
| | 100 | 200 | 300 | 400 | 500 |
| FedAvg | 31.15 | 31.86 | 30.05 | 28.70 | 26.12 |
| FedAvg+ASD (ours) | **37.67** | **35.56** | **32.86** | **29.98** | **27.16** |
| FedDyn | 39.17 | 36.11 | 34.24 | 31.09 | 26.87 |
| FedDyn+ASD (ours) | **39.34** | **40.08** | **36.83** | **33.61** | **28.05** |

Table 10: Impact of increasing the number of clients on the Accuracy (%) when the participation rate is fixed to 2% and **non-iid partitioning of** $\delta = 0.6$.

| Method | Number of Clients | | | | |
|---|---|---|---|---|---|
| | 100 | 200 | 300 | 400 | 500 |
| FedAvg | 35.17 | 32.06 | 30.12 | 28.31 | 25.71 |
| FedAvg+ASD (ours) | **39.61** | **35.49** | **32.16** | **29.43** | **27.61** |
| FedDyn | 37.96 | 36.56 | 34.71 | 30.37 | 26.45 |
| FedDyn+ASD (ours) | **39.40** | **40.49** | **37.48** | **33.23** | **28.60** |

In Table 11, unlike the previous ablation, here we fix the number of clients to 100 and vary the client participation rate from 5%, 10% and 15%. We consider the CIFAR-100 dataset with non-iid partitioning of ($\delta = 0.3$). As expected, the accuracy of the FL-trained models improve with an increase in the client participation rate. We would also like to highlight here that, by adding our proposed ASD strategy consistently improves the accuracy when combined on top of the existing baseline algorithms such as FedAvg and FedDyn.

Table 11: Impact of increasing the client participation rate on the Accuracy (%) with number of clients fixed to 100.

| Method | non iid partition ($\delta = 0.3$) | | | non-iid partition ($\delta = 0.6$) | | |
|---|---|---|---|---|---|---|
| | client paticipation | | | client participation | | |
| | 5% | 10% | 15% | 5% | 10% | 15% |
| FedAvg | 38.22 | 38.67 | 38.85 | 39.04 | 38.53 | 38.00 |
| FedAvg+ASD (Ours) | **43.04** | **42.77** | **43.59** | **43.51** | **42.54** | **42.90** |
| FedDyn | 44.68 | 47.56 | 47.87 | 45.18 | 48.60 | 48.74 |
| FedDyn+ASD (Ours) | **47.51** | **49.03** | **50.32** | **47.81** | **50.23** | **51.48** |

### A.8 Hessian Analysis

In the Table 12 we analyze the top eigenvalue and the trace of the Hessian of the global model when ASD is applied to methods such as FEdProx, FedNTD, FedSAM and FedDisco.

Table 12: The table shows the impact of ASD on the algorithms on CIFAR-100 Dataset. We consistently see that the top eigenvalue and the trace of the Hessian decrease and the Accuracy improves when ASD is used. This suggests that using ASD makes the global model reach to a flat minimum for better generalization.

| Algorithm | Top Eigenvalue ↓ | Trace ↓ | Accuracy ↑ |
|---|---|---|---|
| FedProx | 45.2 | 8683 | 37.79 |
| FedProx + ASD | **11.9** | **2663** | **41.31** |
| FedNTD | **16.3** | 3517 | 40.40 |
| FedNTD + ASD | 17.5 | **2840** | **43.01** |
| FedSAM | 19.04 | 4022 | 40.89 |
| FedSAM + ASD | **6.0** | **1339** | **43.99** |
| FedDisco | 46.7 | 8771 | 38.97 |
| FedDisco + ASD | **12.2** | **2334** | **41.55** |

## A.9 Performance on CIFAR-10 dataset

In Table 13, we show the results for the CIFAR-10 dataset, we find that applying the ASD improves the performance of all the algorithms consistently.

Table 13: We show the accuracy attained by the algorithms on CIFAR-10 at the end of 500 communication rounds. It can be seen that by combining the proposed approach the performance of all the algorithms is improved.

| Algorithm | $\delta = 0.3$ | $\delta = 0.6$ | iid |
|---|---|---|---|
| FedAvg | 78.15 ±0.78 | 78.66 ±0.10 | 80.99 ±0.09 |
| FedAvg+ASD (**Ours**) | **79.01** ±0.33 | **79.93** ±0.21 | **81.83** ±0.19 |
| FedProx | 78.25 ±0.68 | 78.81 ±0.69 | 81.04 ±0.34 |
| FedProx+ASD (**Ours**) | **78.77** ±0.49 | **79.91** ±0.12 | **81.74** ±0.06 |
| FedNTD | 76.79 ±0.37 | 78.55 ±0.31 | 80.98 ±0.21 |
| FedNTD+ASD (**Ours**) | **78.78** ±0.86 | **80.13** ±0.49 | **81.80** ±0.11 |
| FedDyn | 81.08 ±0.52 | 81.48 ±0.35 | 83.51 ±0.27 |
| FedDyn+ASD (**Ours**) | **81.82** ±0.56 | **82.33** ±0.39 | **84.09** ±0.15 |
| FedDisco | 78.21 ±0.45 | 78.76 ±0.32 | 81.04 ±0.30 |
| FedDisco+ASD (**Ours**) | **78.97** ±0.01 | **79.98** ±0.35 | **81.71** ±0.21 |
| FedSpeed | 81.28 ±0.32 | 81.83 ±0.36 | 83.67 ±0.14 |
| FedSpeed+ASD (**Ours**) | **81.70** ±0.20 | **82.62** ±0.26 | **84.57** ±0.24 |

## A.10 Accuracy vs Communication rounds

In the below figures 8 9 and 10, we present how the accuracy is evolving across the communication rounds for the FL methods FedNTD, FedProx, FedDisco with and without the ASD regularizer. We present these results for non-iid ($\delta = 0.3$ and $\delta = 0.6$) and with the iid data partitions for both the CIFAR-100 and Tiny-ImageNet datasets. It can be seen that adding ASD to these off-the-shelf FL methods consistently improves the performance.

## A.11 Privacy of Proposed Method

In our method, which is ASD regularizer, the adaptive weights are computed by the client without depending on the server and it does not assume access to any auxiliary data at the server as assumed in methods such as FedCAD (He et al., 2022b) and FedDF (Lin et al., 2020). In our method, only model parameters are communicated with the server similar to FedAvg (McMahan et al., 2017). Thus our privacy is similar to the FedAvg method at the same time obtaining significant improvements in the performance.

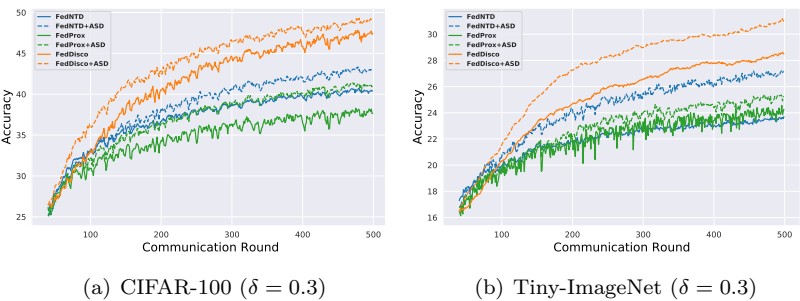

(a) CIFAR-100 ($\delta = 0.3$)  (b) Tiny-ImageNet ($\delta = 0.3$)

Figure 8: Test Accuracy vs Communication rounds: Comparison of algorithms with $\delta = 0.3$, data partition on CIFAR-100 and Tiny-ImageNet datasets. All the algorithms augmented with proposed regularization (ASD) outperform compared to their original form.

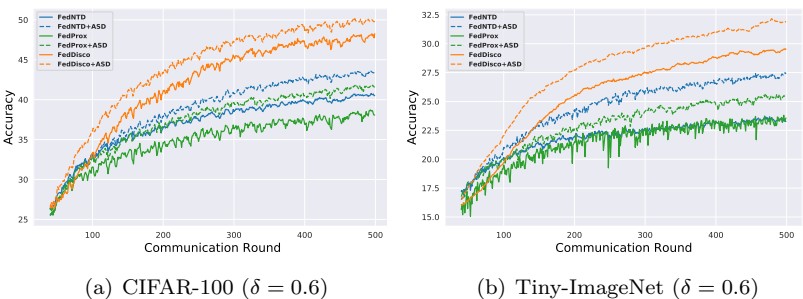

(a) CIFAR-100 ($\delta = 0.6$)  (b) Tiny-ImageNet ($\delta = 0.6$)

Figure 9: Test Accuracy vs Communication rounds: Comparison of algorithms with $\delta = 0.6$ data partition on CIFAR-100 and Tiny-ImageNet datasets. All the algorithms augmented with proposed regularization (ASD) outperform compared to their original form.

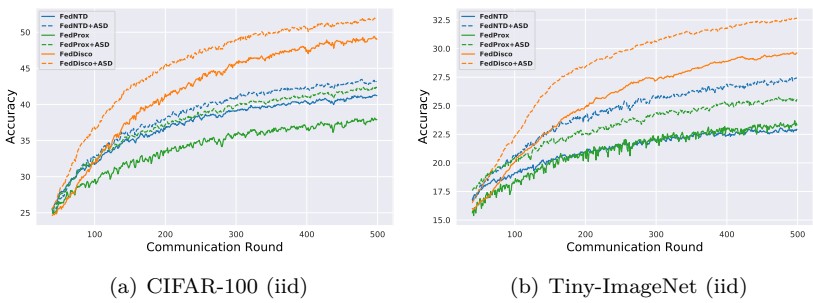

(a) CIFAR-100 (iid)  (b) Tiny-ImageNet (iid)

Figure 10: Test Accuracy vs Communication rounds: Comparison of algorithms with iid data partitions on CIFAR-100 and Tiny-ImageNet datasets. All the algorithms augmented with proposed regularization (ASD) outperform compared to their original form.

### A.12    Implementation of ASD with the FL Methods

.  We now present the integration of ASD loss with the existing FL methods. For all the methods FedAvg, FedDyn, FedSpeed, FedProx, FedDisco and FedSAM, we augment the client loss of each of these methods with our proposed ASD loss in the Eq 5. FedNTD (Lee et al., 2022) uses the non-true distillation loss, it distills the knowledge only from the non-true classes.

$$\mathcal{D}_{\text{NTD}}(q_g(x^i)||q_k(x^i)) = \sum_{c \neq y}^{C} q_g^c(x^i) log(q_g^c(x^i)/q_k^c(x^i)) \tag{17}$$

The above equation represents the FedNTD loss on the sample $i$, when the true class label is $y$. We now use the adaptive weights as defined in Eq. 20, to update the FedNTD loss as below.

$$L_k^{asd-ntd}(\mathbf{w}) \triangleq \sum_{i \in [B]} \alpha_i^k \mathcal{D}_{\text{NTD}}(q_g(x^i)||q_k(x^i)) \tag{18}$$

So the final loss used for optimizing FedNTD with adaptive self-distillation is given below.

$$f_k(\mathbf{w}) \triangleq L_k(\mathbf{w}) + \lambda L_k^{asd-ntd}(\mathbf{w}) \tag{19}$$

where $L_k(\mathbf{w})$ is defined as in Eq. 3 of the main paper.

### A.13    On the choice of KL divergence

The distillation loss introduced by the seminal work of (Hinton et al., 2015) matches the temperature-raised softmax values between the pre-trained teacher model and student model for effective knowledge transfer. It is essentially cross entropy between two softmax vectors. KL divergence differs from cross entropy by a constant and hence achieves the same optimization objective. In our context, we treat the server model as the teacher model and the client model as the student model. Other divergence measures such as reverse KL and JS can also be considered, but we did not see any significant performance difference empirically. In fact KL divergence performed better compared to reverse-KL and JS divergence as shown in Table below. For this experiment we used the CIFAR-100 dataset with 100 clients and 10% client participation rate.

Table 14: Comparison of Statistical Divergences

| Method | Accuracy (in %) |
|---|---|
| KL (ours) | **42.77** |
| reverse-KL | 42.04 |
| Jensen-Shannon | 42.21 |

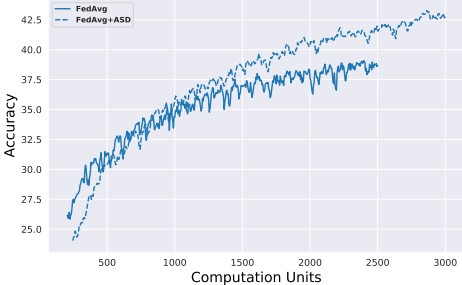

Figure 11: Comparison of the Communication vs Computation for FedAvg and FedAvg+ASD. It can be seen that over the communication rounds for a given amount of computation, FedAvg+ASD attains better accuracy compared to FedAvg.

### A.14   Computation vs Accuracy

In the Figure 11 we have compared the computation with the accuracy for FedAvg and FedAvg+ASD methods. In particular, we observe that at a fixed cumulative computation cost of 2500 units FedAvg attains 38.67 % Accuracy while FedAvg+ASD attains 42.3% accuracy. Here the one unit denotes the computation required for the single forward pass.

### A.15   Proofs of Propositions:

We rewrite the adaptive weighting equations for convenience as below.

$$\alpha_k^i = \frac{\hat{\alpha_k}^i}{\sum_{i \in B} \hat{\alpha_k}^i} \tag{20}$$

and $\hat{\alpha_k}^i$ is defined as below Eq. 21

$$\hat{\alpha_k}^i \triangleq \frac{exp(-\mathcal{H}(x^i))}{p_k^{y^i}} \tag{21}$$

**Proposition A.1.** $\inf_{\mathbf{w} \in \mathbb{R}^d} G_d(\mathbf{w}, \lambda)$ *is* $1, \forall \lambda$

*Proof.*

$$G_d = \frac{\frac{1}{K} \sum_k \|\nabla f_k\|^2}{\|\nabla f\|^2} \tag{22}$$

$$\|\nabla f\|^2 = \|\frac{1}{K} \sum_k \nabla f_k\|^2 \tag{23}$$

We observe that the function $\|.\|^2$ is Convex. By applying Jensen's inequality, we get the desired result. The expectation is taken over the discrete probability measure.

$$\|\nabla f\|^2 \le \frac{1}{K} \sum_k \|\nabla f_k\|^2 \tag{24}$$

$\square$

**Lemma A.2.** *For any function of the form* $\zeta(x) = \frac{ax^2 + bx + c_n}{ax^2 + bx + c_d}$ *satisfying* $c_n > c_d$ *,* $\exists$ $x_c \ge 0$ *such that* $\frac{d\zeta(x)}{dx} < 0 \ \forall \ x \ge x_c$

*Proof.*

$$\frac{d\zeta(x)}{dx} = \frac{(2ax + b)(ax^2 + bx + c_d) - (2ax + b)(ax^2 + bx + c_n)}{(ax^2 + bx + c_d)^2} \tag{25}$$

By re-arranging and simplifying the above we get the following

$$\frac{d\zeta(x)}{dx} = \frac{2x(ac_d - ac_n) + b(c_d - c_n)}{(a_d x^2 + bx + c_d)^2} \tag{26}$$

We are interested in knowing when the numerator is negative.

$$x2a(c_d - c_n) \le b(c_n - c_d) \tag{27}$$

Since $c_n > c_d$, we have

$$x2a(c_n - c_d) \ge -b(c_n - c_d) \implies x \ge \frac{-b}{2a} \tag{28}$$

assuming $x_c = |\frac{-b}{2a}|$

We have the desired condition for $x \geq x_c$

This concludes the proof.

$\square$

**Proposition A.3.** *When the class conditional distribution across the clients is identical, i.e., $\mathbb{P}_k(x \mid y) = \mathbb{P}(x \mid y)$ then $\nabla f_k(\mathbf{w}) = \sum_c p_k^c(\mathbf{g}_c + \lambda \gamma_k^c \tilde{\mathbf{g}}_c)$, where $\mathbf{g}_c = \nabla \mathbb{E}[l(\mathbf{w}; x, y) \mid y = c]$, and $\tilde{\mathbf{g}}_c = \nabla \mathbb{E}[\exp(-\mathcal{H}(x))\mathcal{D}_{KL}(q_g(x)||q_k(x)) \mid y = c]$ where $\gamma_k^c = \frac{1}{p_k^c}$.*

*Proof.* We re-write the equations for $f_k(\mathbf{w})$ from Sec 3.3 of main paper, $L_k(\mathbf{w})$ and $L_k^{ASD}(\mathbf{w})$ from the Sec 3.2 of main paper for convenience.

$$f_k(\mathbf{w}) = L_k(\mathbf{w}) + \lambda L_k^{ASD}(\mathbf{w}) \tag{29}$$

$$L_k(\mathbf{w}) = \underset{x,y \in D_k}{E}[l_k(\mathbf{w}; (x,y))] \tag{30}$$

$$L_k^{ASD}(\mathbf{w}) \triangleq \mathbb{E}[\alpha_k(x,y)\mathcal{D}_{KL}(q_g(x)||q_k(x))] \tag{31}$$

By applying the tower property of expectation, we expand Eq. 30 as below

$$L_k(\mathbf{w}) = \sum_c p_k^c \mathbb{E}[l_k(\mathbf{w}; x, y) \mid y = c] \tag{32}$$

If we assume the class-conditional distribution across the clients to be identical the value of $\mathbb{E}[l_k(\mathbf{w}; x, y) \mid y = c]$ is same for all the clients. Under such assumptions, we can drop the client index $k$ and rewrite the Eq. 32 as follows

$$L_k(\mathbf{w}) = \sum_c p_k^c \mathbb{E}[l(\mathbf{w}; x, y) \mid y = c] \tag{33}$$

$$\nabla L_k(\mathbf{w}) = \sum_c p_k^c \nabla \mathbb{E}[l(\mathbf{w}; x, y) \mid y = c] \tag{34}$$

We further simplify the notation by denoting $\mathbf{g}_c = \nabla \mathbb{E}[l(\mathbf{w}; x, y) \mid y = c]$.

$$\nabla L_k(\mathbf{w}) = \sum_c p_k^c \mathbf{g}_c \tag{35}$$

To make the analysis tractable, In Eq. 21, we use the un-normalized weighting scheme as the constant can be absorbed into $\lambda$. we can re-write Eq. 21 as below

$$\hat{\alpha}_k^i = \gamma_k^y \exp(\mathcal{H}(x)) \tag{36}$$

where $\gamma_k^y = \frac{1}{p_k^y}$ With the above assumptions we can interpret the Eq. 31 as follows.

$$L_k^{ASD} = \underset{x,y \in D_k}{E}[l_k^{dist}(\mathbf{w}; (x,y))] \tag{37}$$

where $l_k^{dist}(\mathbf{w}; (x,y)) = \gamma_k^y \exp(-\mathcal{H}(x))\mathcal{D}_{KL}(q_g(x)||q_k(x))$.

By following the similar line of arguments from Eq. 32 to Eq. 34 we can write the following

$$\nabla L_k^{ASD}(\mathbf{w}) = \sum_c p_k^c \tilde{\mathbf{g}}_c \gamma_k^c \tag{38}$$

$$\nabla f_k(\mathbf{w}) = \sum_c p_k^c (\mathbf{g}_c + \lambda \gamma_k^c \tilde{\mathbf{g}}_c) \tag{39}$$

$\square$

**Lemma A.4.** *If $c_n = K \sum_{k=1}^{K} \sum_{c=1}^{C} (p_k^c)^2$, $c_d = \sum_{k1=1}^{K} \sum_{k2=1}^{K} \sum_{c=1}^{C} (p_{k1}^c p_{k2}^c)$. where $p_k^c \geq 0 \ \forall k, c$, and $\sum_{c=1}^{C} p_k^c = 1$, then $\frac{c_n}{c_d} \geq 1$.*

*Proof.* We need to show that

$$\frac{\sum_{k=1}^{K} \sum_{c=1}^{C} (p_k^c)^2}{\sum_{k1=1}^{K} \sum_{k2=1}^{K} \sum_{c=1}^{C} (p_{k1}^c p_{k2}^c)} \geq \frac{1}{K} \tag{40}$$

By rewriting the denominator we get

$$\frac{\sum_{k=1}^{K} \sum_{c=1}^{C} (p_k^c)^2}{\sum_{c=1}^{C} (\sum_{k=1}^{K} (p_k^c))^2} \geq \frac{1}{K} \tag{41}$$

Consider rewriting the denominator of the L.H.S of above equation.

$$\sum_{c=1}^{C} (\sum_{k=1}^{K} (p_k^c))^2 = \sum_{c=1}^{C} ((\mathbf{p^c})^{\intercal} \mathbf{1})^2 \tag{42}$$

where $\mathbf{p^c} = [p_1^c p_2^c ... p_K^c]^{\intercal}$ and $\mathbf{1}$ is the all one vector of size $K$

Applying the Cauchy Schwartz inequality to the R.H.S of the Eq. 42 we get the following.

$$\sum_{c=1}^{C} ((\mathbf{p^c})^{\intercal} \mathbf{1})^2 \leq \sum_{c=1}^{C} \sum_{k=1}^{K} (p_k^c)^2 K \tag{43}$$

By combining the Eq. 42 and Eq. 43 the result follows. $\square$

**Proposition A.5.** *When the class-conditional distribution across the clients is the same, and the Assumption 3.3 holds then $\exists$ a range of values for $\lambda$ such that whenever $\lambda \geq \lambda_c$ we have $\frac{dG_d}{d\lambda} < 0$ and $G_d(\mathbf{w}, \lambda) < G_d(\mathbf{w}, 0)$.*

*Proof.*

$$G_d = \frac{\frac{1}{K} \sum_{k=1}^{K} \|\nabla f_k\|^2}{\|\nabla f\|^2} \tag{44}$$

From Sec 3.2 of the main paper we have the following, We drop the argument $\mathbf{w}$ for the functions $f_k$ to simplify the notation

$$\nabla f_k = \sum_{c=1}^{C} p_k^c (g_c + \lambda \gamma_k^c \tilde{g}_c) \tag{45}$$

$$\|\nabla f_k\|^2 = \sum_{c1=1}^{C} \sum_{c2=1}^{C} p_k^{c1} p_k^{c2} (\mathbf{g}_{c1}^{\mathsf{T}} + \lambda \gamma_k^{c1} \tilde{\mathbf{g}}_{c1}^{\mathsf{T}})(\mathbf{g}_{c2} + \lambda \gamma_k^{c2} \tilde{\mathbf{g}}_{c2})$$

$$= \sum_{c1=1}^{C} \sum_{c2=1}^{C} p_k^{c1} p_k^{c2} (\mathbf{g}_{c1}^{\mathsf{T}} \mathbf{g}_{c2} + \lambda \gamma_k^{c2} \mathbf{g}_{c1}^{\mathsf{T}} \tilde{\mathbf{g}}_{c2} + \lambda \gamma_k^{c1} \tilde{\mathbf{g}}_{c1}^{\mathsf{T}} \mathbf{g}_{c2} + \lambda^2 \gamma_k^{c2} \gamma_k^{c1} \tilde{\mathbf{g}}_{c1}^{\mathsf{T}} \tilde{\mathbf{g}}_{c2})$$

$$\approx \sum_{c=1}^{C} (p_k^c)^2 (\mathbf{g}_c^{\mathsf{T}} \mathbf{g}_c) + \lambda \sum_{c1=1}^{C} \sum_{c2=1}^{C} p_k^{c1} \mathbf{g}_{c1}^{\mathsf{T}} \tilde{\mathbf{g}}_{c2} + \lambda \sum_{c1=1}^{C} \sum_{c2=1}^{C} p_k^{c2} \tilde{\mathbf{g}}_{c1}^{\mathsf{T}} \mathbf{g}_{c2} + \lambda^2 \sum_{c=1}^{C} (p_k^c)^2 \gamma_k^c \gamma_k^c \tilde{\mathbf{g}}_c^{\mathsf{T}} \tilde{\mathbf{g}}_c$$

$$= \sum_{c=1}^{C} (p_k^c)^2 + 2\lambda \sum_{c1=1}^{C} \sum_{c2=1}^{C} p_k^{c1} \mathbf{g}_{c1}^{\mathsf{T}} \tilde{\mathbf{g}}_{c2} + \lambda^2 C \tag{46}$$

In the above equation the second equality is obtained by simply expanding the product, the third approximation by weakly correlated assumption of the gradients. The last two equalities used the fact that $\gamma_k^c = \frac{1}{p_k^c}$. We also assume that gradients are normalized to unit magnitude.

Finally, we have the following

$$\frac{1}{K} \sum_{k=1}^{K} \|\nabla f_k\|^2 = \frac{1}{K} \Big( \sum_{k=1}^{K} \sum_{c=1}^{C} (p_k^c)^2 + 2\lambda \sum_{k=1}^{K} \sum_{c1=1}^{C} \sum_{c2=1}^{C} p_k^{c1} \mathbf{g}_{c1}^{\mathsf{T}} \tilde{\mathbf{g}}_{c2} + \lambda^2 KC \Big) \tag{47}$$

$$= \frac{1}{K^2} (a_n \lambda^2 + b_n \lambda + c_n)$$

where

$$a_n := K^2 C \tag{48}$$

$$b_n := 2K \sum_{k=1}^{K} \sum_{c1=1}^{C} \sum_{c2=1}^{C} p_k^{c1} \mathbf{g}_{c1}^{\mathsf{T}} \tilde{\mathbf{g}}_{c2}$$

$$c_n := K \sum_{k=1}^{K} \sum_{c=1}^{C} (p_k^c)^2 \tag{49}$$

$$\|\nabla f\|^2 = \Big( \Big\| \frac{1}{K} \sum_{k=1}^{K} \sum_{c=1}^{C} p_k^c (\mathbf{g}_c + \lambda \gamma_k^c \tilde{\mathbf{g}}_c) \Big\| \Big)^2$$

$$= \frac{1}{K^2} \sum_{k1=1}^{K} \sum_{k2=1}^{K} \sum_{c1=1}^{C} \sum_{c2=1}^{C} p_{k1}^{c1} (\mathbf{g}_{c1}^{\mathsf{T}} + \lambda \gamma_{k1}^{c1} \tilde{\mathbf{g}}_{c1}^{\mathsf{T}}) p_{k2}^{c2} (\mathbf{g}_{c2} + \lambda \gamma_{k2}^{c2} \tilde{\mathbf{g}}_{c2})$$

$$\approx \frac{1}{K^2} \sum_{k1=1}^{K} \sum_{k2=1}^{K} \Big( \sum_{c=1}^{C} p_{k1}^c p_{k2}^c (\mathbf{g}_c^{\mathsf{T}} \mathbf{g}_c)$$

$$+ \sum_{c1=1}^{C} \sum_{c2=1}^{C} \big( \lambda p_{k1}^{c1} p_{k2}^{c2} \gamma_{k2}^{c2} \mathbf{g}_{c1}^{\mathsf{T}} \tilde{\mathbf{g}}_{c2} + \lambda p_{k1}^{c1} p_{k2}^{c2} \gamma_{k2}^c \tilde{\mathbf{g}}_{c1}^{\mathsf{T}} \mathbf{g}_{c2} + \lambda^2 p_{k1}^{c1} p_{k2}^{c2} \gamma_{k1}^c \gamma_{k2}^c \tilde{\mathbf{g}}_c^{\mathsf{T}} \tilde{\mathbf{g}}_c \big) \Big)$$

$$= \frac{1}{K^2} \sum_{k1=1}^{K} \sum_{k2=1}^{K} \sum_{c=1}^{C} (p_{k1}^c p_{k2}^c) + \lambda \sum_{k1=1}^{K} \sum_{k2=1}^{K} \sum_{c1=1}^{C} \sum_{c2=1}^{C} p_{k1}^{c1} \mathbf{g}_{c1}^{\mathsf{T}} \tilde{\mathbf{g}}_{c2} + \lambda \sum_{k1=1}^{K} \sum_{k2=1}^{K} \sum_{c1=1}^{C} \sum_{c2=1}^{C} p_{k2}^{c2} \tilde{\mathbf{g}}_{c1}^{\mathsf{T}} \mathbf{g}_{c2} + \lambda^2 K^2 C$$

$$= \frac{1}{K^2} \Big( \sum_{k1=1}^{K} \sum_{k2=1}^{K} \sum_{c=1}^{C} (p_{k1}^c p_{k2}^c) + 2\lambda K \sum_{k=1}^{K} \sum_{c1=1}^{C} \sum_{c2=1}^{C} p_k^{c1} \mathbf{g}_{c1}^{\mathsf{T}} \tilde{\mathbf{g}}_{c2} + \lambda^2 K^2 C \Big)$$

$$= \frac{1}{K^2} (a_d \lambda^2 + b \lambda + c_d) \tag{50}$$

By defining

$$a_d := K^2 C \tag{51}$$

$$b_d := 2K \sum_{k=1}^{K} \sum_{c1=1}^{C} \sum_{c2=1}^{C} p_k^{c1} \mathbf{g}_{c1}^{\mathsf{T}} \tilde{\mathbf{g}}_{c2}$$

$$c_d := \sum_{k1=1}^{K} \sum_{k2=1}^{K} \sum_{c=1}^{C} (p_{k1}^c p_{k2}^c) \tag{52}$$

By substituting Eq. 47 and Eq. 50 in Eq. 44 we get

$$G_d(\mathbf{w}, \lambda) = \frac{a_n \lambda^2 + b_n \lambda + c_n}{a_d \lambda^2 + b_d \lambda + c_d} \tag{53}$$

Comparing Eq. 48 and Eq. 51 we see that $a := a_n = a_d$, $b := b_n = b_d$.

Also $c_n > c_d$ assuming $p_k^c$ is non-degenerate.

Using the Lemma A.2 on Eq. 53 we get the value of $\lambda_b$ such that $G_d(\mathbf{w}, \lambda)$ is reduced.

We also get $\lambda \geq |\frac{-b}{a}|$ by analyzing the values of $\lambda$ for which $G_d(\mathbf{w}, \lambda) < G_d(\mathbf{w}, 0)$ holds.

Thus choosing the $\lambda > \lambda_c = \sup_{b: -k^2 C \leq b \leq k^2 C} max(\lambda_b, |\frac{-b}{a}|)$ guarantees $G_d(\mathbf{w}, \lambda) < G_d(\mathbf{w}, 0)$, for all $\mathbf{w}$.

This concludes the proof.

$\square$

## A.16 Discussion on Impact of Gradient Dissimilarity on the Convergence

We now study how the gradient diversity impacts the convergence of the FL algorithms such as FedProx and FedAvg. We omit the dependence of $\lambda$ on $B$. (for these algorithms $\lambda = 0$ so $B$ is nothing but $B(0)$ in our notation) We have the gradient dissimilarity assumption below

**Assumption A.6.** $\frac{1}{K} \sum_k \|\nabla f_k(\mathbf{w})\|^2 \leq B^2 \|\nabla f(\mathbf{w})\|^2$

### A.16.1 FedProx

Suppose the functions $f_k$ are lipschiltz smooth and their exists $L_- > 0$ such that $\mathbf{H} f_k \succeq L_- \mathbf{I}$. With $\bar{\mu} - L > 0$, where $\mu$ is FedProx regularization. If $f_k$ satisfies the assumption A.6 then acccording to Theorem 6 of (Li et al., 2020) the FedProx, after $T = \mathcal{O}(\frac{\Delta}{\rho \epsilon})$. We have the gradient contraction as $\frac{1}{T} \sum_{t=0}^{T-1} \mathbf{E} \|f(\mathbf{w}^t)\|^2 \leq \epsilon$. The value of $\rho$ is given below.

$$\rho = \frac{1}{\mu} - \frac{\gamma B}{\mu} - \frac{B(1+\gamma)\sqrt{(2)}}{\bar{\mu}\mu} - \frac{L(1+\gamma)^2 B^2}{2\bar{\mu}^2} - \frac{L(1+\gamma)^2 B^2}{K\bar{\mu}^2}(2\sqrt{2K} + 2) > 0 \tag{54}$$

for some $\gamma > 0$ and $\Delta = f(\mathbf{w}^0) - f(\mathbf{w}^*)$, $f(\mathbf{w}^*)$ is the local minimum.
It can be seen that the convergence is inversely related to $\rho$. High value of $\rho$ leads to faster convergence. From Eq. 54 we can see that $\rho$ can be increased by decreasing the value of $B$. Thus reducing the value of $B$ helps in better convergence.

### A.16.2 FedAvg

**Assumption A.7.** We now analyze the convergence of FedAvg, we consider the following assumptions $\|\nabla f_k(\mathbf{x}) - f_k(\mathbf{x})\| = \beta\|\mathbf{x} - \mathbf{y}\|$ ($\beta$ smoothness)

**Assumption A.8.** Gradients have bounded Variance.

Suppose that $f(\mathbf{w})$ and $f_k(\mathbf{w})$, satisfies Assumptions A.6, A.7 and A.8. Let $\mathbf{w}^* = \underset{w}{arg\,min}\ f(\mathbf{w})$ the local step-size be $\alpha_l$. The theorem V in (Karimireddy et al., 2020) shows that FedAvg algorithm will have contracting gradients. If Initial model is $\mathbf{w}^0$, $F = f(\mathbf{w}^0) - f(\mathbf{w}^*)$ and for constant $M$, then in $R$ rounds, the model $w^R$ satisfies $\mathbb{E}[\|\nabla f(\mathbf{w}^R)\|^2] \leq O(\frac{\beta M\sqrt{F}}{\sqrt{RLS}} + \frac{\beta B^2 F}{R})$.

We see the convergence rate is $O(\frac{\beta M\sqrt{F}}{\sqrt{RLS}} + \frac{\beta B^2 F}{R})$. We can see that convergence has a direct dependence on $B^2$. This is the only term that is linked to heterogeneity assumption. So the lower value of $B$ implies faster convergence. This motivates to have a tighter bound on heterogeneity. ASD achieves this by introducing the regularizer and choosing the appropriate value of $\lambda$. In the figure 12 we empirically we verify the impact of ASD on the convergence. We plot the smoothed estimates of the norm of the difference of the global model parameters between the successive communication rounds i.e $\|\mathbf{w}^t - \mathbf{w}^{t-1}\|$.

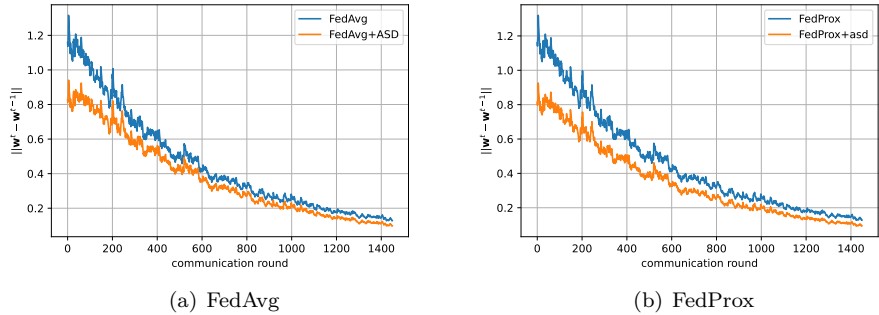

(a) FedAvg  (b) FedProx

Figure 12: Impact of ASD on the convergence on CIFAR-100 dataset with non-iid partition of $\delta = 0.3$

