# OpenReview forum: "Adaptive Self-Distillation for Minimizing Client Drift in Heterogeneous Federated Learning"
_TMLR — Accepted by TMLR_

### Review · Reviewer_SCnE · 2024-06-20

**Summary Of Contributions:**

This paper considered heterogeneity federated learning, where the label distributions changes significantly across the clients. This paper proposed a specific regularization ASD - Adaptive Self-Distillation. The proposed regularization is a weighted KL divergence to ensure the local update on clients similar to the global model.

**Audience:**

Yes

**Claims And Evidence:**

Yes

**Requested Changes:**

I would suggest authors to revised the following parts

- The introduction. Current introduction seems a bit convoluted by previous work and the actual contribution. I would suggest to briefly discuss the most important points.

- Please address the concerns on the choice of $\alpha$.

- Please address the concerns on the choice of statistical divergence.

**Strengths And Weaknesses:**

Strong points

- The contexts and background are clearly presented. Indeed I am not an expert in FL and just familiar with some well-known papers. The problem setup, introduction and proposed losses are clearly illustrated.
- The motivation is clearly illustrated. Indeed, Fig 1 clearly demonstrates the limitations of other methods. Then naturally introducing the proposed method. I appreciate this narrative.
- The experiments seem strong. The empirical results seem significant.

Weak points

- In the introduction. Presenting other’s related work seems a bit convoluted. In general, this will render the paper a bit inaccessible.
- In the proposed losses. I could understand the importance of weights. But I could not understand why $\alpha$ is defined in this manner.
- Why KL divergence in the distillation is preferred? Why not other divergences? I feel a bit confused about the rationale of divergence choice.

Overall I feel like this paper is worthy for publication. But I feel like some parts are still lacking convincing points to justify.

---

> ### Author Response · Authors · 2024-08-29
> **Response to the reviewer  SCnE**
>
> We thank the reviewer for reviewing our work and providing suggestions for improving the draft. We now address the queries raised by the reviewer.
>
> **Changes suggested in the Introduction:**
>
> We thank the reviewer for the suggestion.  As suggested by the reviewer, we have modified the introduction section in the revised draft by retaining only the relevant previous work and moved discussions about other works to the related works section.
>
> **On the choice of**  $ \alpha $
>
> Our ASD scheme mitigates client drift by ensuring close alignment between the global and individual client models. This alignment is achieved via self-distillation employing adaptive weights ($\alpha$), which are dynamically adjusted based on the global model's prediction entropy and the empirical label distribution of the training data on the client.
>
> The choice of $\alpha$ is designed to ensure that, when the global model encounters samples with high prediction entropy, we decrease the weighting for the regularization loss. Conversely, for samples with a low probability of occurrence, we prioritize learning from the global model. This adaptive approach enables local models to effectively learn from the cross-entropy loss for data samples from the classes that are more represented in the training set for that client while leveraging the global model's guidance for less frequent labels. In Table 3 of the paper, we highlight the importance of adaptive weights, where we clearly show that ASD with adaptive weights consistently improves performance when combined with off-the-shelf FL methods. We have clarified this in the Sec 3.2 of the revised manuscript.
>
> **Regarding the choice of statistical divergence**
>
> The distillation loss introduced by the seminal work of Hinton et.al.[1] matches the temperature-raised softmax values between the pre-trained teacher and student models for effective knowledge transfer. It is essentially cross entropy between two softmax vectors. Kullback–Leibler (KL) divergence differs from cross entropy by a constant and hence achieves the same optimization objective. In our context, we treat the server model as the teacher model and the client model as the student model. Other divergence measures such as reverse KL and Jensen-Shannon divergence (JS) can also be considered, but we did not see any significant performance difference empirically. In fact, KL divergence performed better compared to reverse KL and JS divergence in our experiments. For example, the following table presents the results for the CIFAR-100 dataset with different choices of divergences. We consider the Dirichlet non-iid (0.3) based data partitioning across the clients, and the accuracies are reported at the end of 500 communication rounds.
>
> | Divergence Method | Accuracy (\%)|
> |----------------------- |-----------------|
> | KL                                 | 42.77             |
> | reverse KL                     |  42.04            |
> | Jensen Shannon (JS)    |   42.21           |
>
>  We have added this experiment in Sec A.13 of the revised manuscript.
>
> [1] Distilling the Knowledge in a Neural Network, Hintion et.al.

---

### Review · Reviewer_zf56 · 2024-07-01

**Summary Of Contributions:**

This paper proposes Adaptive Self-Distillation (ASD), a plug-and-play method to be used on top of any FL algorithm to address the client drift problem in Federated Learning (FL).

The technique adds a regularization term to the local loss function to penalize the mismatch between the local model's softmax predictions and the global model's softmax predictions using the Kullback-Leibler divergence. The Self-Distillation is *Adaptive* in that the contributions to the novel loss term are of different intensity depending on the specific sample, controlled by adaptive weights $\alpha_k$, which are functions of the entropy of the global model and the frequency of the class the sample belongs to. The ASD loss is constructed so that $\alpha_k$ is low when the server predictions are noisy, giving more importance to the local predictions and vice versa.

ASD is mathematically proven to reduce client drift according to the gradient dissimilarity measure used in [1, 2]. Moreover, this paper discusses flat minima by analyzing the top eigenvalue and the trace of the Hessian of the training loss. It also provides empirical evidence of how applying ASD ensures convergence to flat minima, guaranteeing better generalization properties.

[1] Li, Tian, et al. "Federated optimization in heterogeneous networks." Proceedings of Machine learning and systems 2 (2020): 429-450.

[2] Lee, Gihun, et al. "Preservation of the global knowledge by not-true distillation in federated learning." Advances in Neural Information Processing Systems 35 (2022): 38461-38474.

**Audience:**

Yes

**Broader Impact Concerns:**

I have no concerns about the ethical impacts of the work.

**Claims And Evidence:**

Yes

**Requested Changes:**

This work presents no critical issues. However, some suggestions to improve its quality are detailed below.

- In the introduction, page 2, the paper states that "the primary motivation behind distillation and regularization works in the context of FL is that the global model will have better representation than the global models". The authors do not justify this statement, nor are some references provided. Please consider clarifying this aspect.
- In the introduction, page 2, the authors should consider discussing more related client drift works or, even better, including a subsection on addressing client drift in Related Works, as ASD is designed to reduce client drift specifically. Some possible citations are [4-7].
- Page 8, it is unclear why $f = \sum_{i=1}^K f_i \implies \mu_1(H(f)) \leq \frac{1}{K} \sum_{i=1}^K \mu_1(H(f_i))$.
- The authors may consider adding results using the datasets introduced in [8] for cross-device FL.
- Line plots could be improved for visibility. For instance, the authors may consider pairing the lines associated with the same method with a similar color (e.g., FedAvg blue, FedAvg + ASD light blue) and/or with different line styles (e.g. [method]: solid, [method] + ASD: dashed).
- The authors may consider including plots to study the required computation for the empirical experiments (x-axis: average computation required per client; y-axis: accuracy).
- Proposition A.1.: I have a doubt about a possible missing assumption for this proposition to hold. The gradient must be a convex function to apply the Jensen's inequality, but $\nabla f$ is convex if $H(f)$ is positive semi-definite, so I think that there is a missing assumption on $H(f)$.
- Lemma A.4.: It is unclear why the R.H.S of Eq. 40 is $\frac{1}{K}$ and not $1$.

[4] Wang, Jianyu, et al. "Slowmo: Improving communication-efficient distributed sgd with slow momentum." ICLR 2020.

[5] Reddi, Sashank, et al. "Adaptive federated optimization." ICLR 2021.

[6] Hsu, Tzu-Ming Harry, Hang Qi, and Matthew Brown. "Measuring the effects of non-identical data distribution for federated visual classification."

[7] Kim, Geeho, Jinkyu Kim, and Bohyung Han. "Communication-efficient federated learning with accelerated client gradient." CVPR 2024.

[8] Hsu, Tzu-Ming Harry, Hang Qi, and Matthew Brown. "Federated visual classification with real-world data distribution." ECCV 2020.

**Strengths And Weaknesses:**

**Strengths**

- The paper is well-written and easy to understand.
- The method is simple yet effective, theoretically proven, and empirically demonstrated.
- The discussion is thorough, and the authors' claims are justified and well-addressed.
- ASD shows improved performance with any algorithm, showing its effectiveness independently from the algorithm and with different tasks.

**Weaknesses**

- The setting where ASD could shine is not thoroughly discussed. The experiments look closer to the cross-silo FL scenario, as there are only 100 clients and a high participation rate (the paper reports 10%), but statistical heterogeneity and client drift are an issue for cross-device FL as well [3].

[3] Kairouz, Peter, et al. "Advances and open problems in federated learning." Foundations and trends® in machine learning 14.1–2 (2021): 1-210.

---

> ### Author Response · Authors · 2024-08-29
> **Response to the reviewer zf56**
>
> We thank the reviewer for reviewing our work and providing feedback. We now address the queries raised by the reviewer.
>
> **Cross Device setting with more than 100 clients**
>
> To mimic the cross-device setting we have considered the CIFAR-100 dataset with the data partitioned among the 500 clients in the non-iid fashion by using the Dirichlet(0.3). We consider only 1\% of the clients participate in every round. We report the accuracy at the end of 1000 rounds.
>
> | Method | Accuracy (\%)|
> |-----------------------|-----------------|
> | FedAvg                   | 27.92             |
> | FedAvg+ASD(ours) | **31.28**              |
> | FedProx                   | 28.09             |
> | FedProx+ASD(ours) | **32.13**              |
> | FedDyn                   | 31.00             |
> | FedDyn+ASD(ours) | **33.12**              |
> | FedNTD                   | 30.09             |
> | FedNTD+ASD(ours) | **33.65**              |
> | FedSpeed                   | 34.08             |
> | FedSpeed+ASD(ours) | **36.59**              |
>
> It is evident from the above table that adding our proposed regularizer ASD on top of existing methods yields consistent improvements even when the number of clients are increased and with lower client participation rate indicating the efficacy of our method. We have added the above experiment in Sec 5.3 of the paper.
>
>
> **Regarding the primary motivation behind distillation and regularization works in the context of FL is that the global model will have better representation than the global models**
>
> We assume the reviewer meant “global model will have better representation than local models”.  By this, we meant that local model overfits its local data due to heterogeneity. Although we initialize the client models with the aggregated global model at each FL round, the subsequent local epochs of training on non IID client data often leads to 'drift' of the client models to their respective local minimum. To mitigate this drift regularization approaches are introduced. We have explained this in the introduction section of the updated submission.
>
> **In the introduction, page 2, the authors should consider discussing more related client drift works or, even better, including a subsection on addressing client drift in Related Works, as ASD is designed to reduce client drift specifically. Some possible citations are [4-7].**
>
> Thank you for the suggestion. We have included section 2.2 on client-drift under the related works.
>
> **Why** $f =\frac{1}{K} \sum_{i=1}^{K}f_i \implies  \mu_{1} (H(f)) \leq \frac{1}{K}  \sum_{i=1}^{K}   \mu_1(H(f_i))$
>
> Since we have $f =\frac{1}{K} \sum_{i=1}^{K}f_i$
>
> It holds that  $H(f) = \frac{1}{K} \sum_{i=1}^{K} H(f_i)$
>
> We assume that Hessian's of each client's loss function exists and is continuous. This implies that each of $H(f_i)$ is symmetric.  We will clearly state this assumption in the paper.
>
> For any symmetric matrices $A$ and $B$ we have the identity $\mu_{1}(A+B)\leq \mu_{1}(A) + \mu_{1}(B)$. Here $\mu_1$ denotes the top eigenvalue. By applying this identity to $H(f)$ above we get the desired inequality below
>
> $\mu_{1} (H(f)) \leq \frac{1}{K}  \sum_{i=1}^{K}   \mu_1(H(f_i))$
>
> **Improving the line plots for better visibility**
>
> As suggested, we have added the line plots for the base algorithms such as FedAvg, FedDyn etc and the dashed plots of the same color for FedAvg+ASD and FedDyn+ASD.
>
> **Computation vs. Accuracy**
>
> We have calculated the computation required for one round of  FL and then plotted the accuracy vs. computation achieved by FedAvg and Fedavg+ASD. In particular, we observe that at a fixed computation cost, say 2500 units, FedAvg attains 38.67 % accuracy while FedAvg+ASD attains 42.3% accuracy. The number 2500 denotes the cumulative number across the communication rounds. The unit is the computation required for the single forward pass.  We have added this discussion in Sec A.14 and Figure 11 of the revised manuscript.
>
> **On proposition A.1**
>
> Jensen's inequality implies that if $f$ is the convex function, we have $\mathbb{E}(f(x)) \geq f(\mathbb{E}(x))$.
>
> The function under consideration here is square of the norm i.e. $|| . ||^2$ which is a convex function. Applying the Jensens inequality to the gradient vectors gives the following desired inequality.
>
> $\frac{1}{K} \sum_{i=1}^{K}|| \nabla{f_i}||^2 \geq ||  \frac{1}{K} \sum_{i=1}^{K} \nabla{f_i} ||^2$
>
> **Lemma A.4.: It is unclear why the R.H.S of Eq. 40 is $\frac{1}{K}$ and not $1$**
>
> We apologize for this typo. In the equation of $c_n$, the scale of K was missing. It is supposed to be $c_n = \{K} \sum_{k=1}^{K}\sum_{c=1}^{C}({p_k^c})^2$. Thank you very much for pointing this out.

---

### Review · Reviewer_J9kY · 2024-08-25

**Summary Of Contributions:**

This paper proposes an algorithm for better federated learning, especially in the setting of non IID clients where the learning must adapt to different label distributions. More specifically, the algorithm is an adaptive self-distillation regularization loss that encourages the models to rely more on the local nodes when there is sufficient labels and data points, while relying more on the global model for sparse labels. Additionally, the paper discusses some theoretical results regarding the gradients of the model, as well as how to use statistics of the Hessian matrix of the loss function estimated on the training data to determine flatness of the minima, and thus generalization. The experiments are on CIFAR-10, CIFAR-100, and Tiny ImageNet, using mostly shallow architectures.

**Audience:**

Yes

**Claims And Evidence:**

Yes

**Requested Changes:**

See above. I would the very least expect

- more extensive ablations on variuos hyperparameters and under different experimental settings and datasets
- try different and bigger datasets, including at the very least ImageNet, which by now is not even considered such a big dataset
- try more modern and high-performing architectures, mainly Transformer-based ones.

**Strengths And Weaknesses:**

Strengths

+ The paper presents a simple, intuitive, and easy-to-understand algorithm.
+ The algorithm has some theoretical justification.
+ Results on the presented experiments are promising. The method is presented as a drop-in replacement to other methods and there seems to be consistent improvement of at least 1-3%, where the bigger improvements are with simpler baseline models and architectures.
+ The paper also features nice visuals, figures, and explanations, making it easy to understand where it works and where it doesn't.

Weaknesses

- While the paper seems simple, the devil is in the details, which in this case is the hyperparameters and the optimization. To me, it is clear and logical to drop back to the global model in the absence of labeled data for training. However, the key question is what comprises sparse data and labels, when to decide to start relying on the global model more, what happens if suddenly more labelled data kick in making the client not sparsely labelled, etc? How sensitive is the algorithm to these hyperparameter choices?
- More generally, I would expect a more thorough ablation experimentation on hyperparameters and experimental settings, including the value lambda, the size of the dataset overall as well as per different server-client ratios, temperature tau, different IID settings or ways of distributing labels, with larger datasets, and more modern and expressive architectures, eg Transformer based ones.
- It has been a while since I worked with federated learning, but from what I know, optimization has always been the major problem. It is not entirely clear how is the presented algorithm is supposed to address problems with the optimization? What if the statistics of the server and client change over time, such that it is unclear when to switch from the global to the local model and vice versa?

---

> ### Author Response · Authors · 2024-08-29
> **Response to reviewer J9kY**
>
> We thank the reviewer for reviewing our work and providing feedback. We are currently performing the suggested experiments and we will update once the results are available.

---

> ### Author Response · Authors · 2024-09-07
> **Response to Reviewer J9kY (1/3)**
>
> **(1) To me, it is clear and logical to drop back to the global model in the absence of labelled data for training. However, the key question is what comprises sparse data and labels, when to decide to start relying on the global model more, what happens if suddenly more labelled data kick in making the client not sparsely labelled, etc? How sensitive is the algorithm to these hyperparameter choices?**
>
> We would like to provide clarification on our FL setup. Similar to [FedDyn], [FedNTD] etc. we assume fully labelled and a fixed set of training data across all the clients. However, the label distributions for training data are non IID across the clients, i.e., these distributions vary from one client to another. In other words,  in any given client labelled data samples for certain classes are more abundant than the rest of the classes and these sets of well-represented classes would vary from client to client. By sparse labels, we mean those labels with a smaller number of representative samples in the dataset.
>
> Note that, in each FL round, the global model is, in fact, an aggregation of the models trained on clients' local data, which is then pushed back to each client for further local updates. To mitigate client drift due to heterogeneity in label distributions across clients, we utilize the global model as the regularizer in an adaptive manner by the weights ($\alpha$). These weights are designed to ensure that, when the global model encounters samples with high prediction entropy, we decrease the weighting on the regularization loss. Conversely, for samples with a low probability of occurrence, we prioritize learning from the global model. This adaptive approach, which is one of the key contributions of this work,   enables local models to effectively learn from the cross-entropy loss for data samples from the classes that are more represented in the training set for that client while leveraging the global model's guidance for less frequent labels.
> Following the reviewer’s suggestion, we have performed a detailed analysis on the choice of different hyper-parameters, which are described in the following responses as well as in Sec A.5, A.6, and A.7  of the paper.
>
> [FedDyn] Federated Learning Based on Dynamic Regularization, ICLR,2021
>
> [FedNTD] Preservation of the Global Knowledge by Not-True Distillation in Federated Learning (NeurIPS,2022)
>
> **(2) More extensive ablations on variuos hyperparameters and under different experimental settings and datasets**
>
> (a) **Impact on the choice of $\lambda$ and $\tau$**
>
> We study the impact of changing the hyper-parameters $\lambda$ and $\tau$ on the CIFAR-100 dataset with the Dirichlet non-iid partition of $\delta = 0.3$. We report the accuracy at the end of 500 rounds, when using FedAvg+ASD algorithm. In Figure 7,  we see that the accuracy of the model increases with $\lambda$ and then slightly drops after certain point. This is expected as too less value of $\lambda$ is similar to FedAvg and very high value of $\lambda$ will ignore the local learning. It can also be seen that for all the values of $\lambda$ the accuracy peaks at $\tau = 2.0$. In all of our experiments we set the temperature parameter $\tau$ to 2.0. The details about these experiments are added to Sec.A.6 of the paper.

---

> ### Author Response · Authors · 2024-09-07
> **Response to Reviewer J9kY (2/3)**
>
> (b) **Impact of increasing the number of clients on the Accuracy (\%) when the participation rate is fixed to 2\%.**
>
> We perform this ablation using the CIFAR-100 dataset with a non-iid Dirichlet data partitioning of $\delta = 0.3$ and $\delta = 0.6$ We fix the client participation to 2\% and vary the number of clients. We summarize our observations in the Table below.
>
> We report the accuracy by varying the clients from 100 to 500 and fixing the client participation to 2% with
>
> $\delta = 0.3$
>
> | Method     |       100 |       200 |       300 |       400 |       500 |
> |------------|----------:|----------:|----------:|----------:|----------:|
> | FedAvg     |      31.5 |     31.86 |     30.05 |      28.7 |     26.12 |
> | FedAvg+ASD (Ours) | **37.67** | **35.56** | **32.86** | **29.98** | **27.16** |
> | FedDyn     |     36.11 |     36.26 |     34.24 |     31.09 |     26.87 |
> | FedDyn+ASD (Ours) | **39.34** | **40.08** | **36.43** | **33.61** | **28.05** |
>
> $\delta = 0.6$
>
> | Method    |   100 |   200 |   300 |   400 |   500 |
> |------------|------:|------:|------:|------:|------:|
> | FedAvg     | 35.17 | 32.06 | 30.12 | 28.31 | 25.71 |
> | FedAvg+ASD (Ours) | **39.61** | **35.49** | **32.16** | **29.43** | **27.61** |
> | FedDyn     | 37.96 | 36.56 | 34.71 | 30.37 | 26.45 |
> | FedDyn+ASD  (Ours) |  **39.40** | **40.49** | **37.48** | **33.23** |  **28.60** |
>
> It can be seen that ASD improves the performance of the baselines FedAvg and FedDyn in all the settings. In particular, we would like to highlight the point here that despite increasing the number of clients, the total number of training data samples across all the clients remains constant (for CIFAR-100). Thus as the number of clients increases, the number of data samples per client decreases . This further aggravates the adverse impact of label heterogeneity across clients, and hence, accuracy degrades in general.
> However, we are happy to observe and report that, even under such a challenging setup, our proposed adaptive self-distillation-based strategy consistently improves the accuracy when combined on top of the existing baseline algorithms. We have also highlighted this observation in the revised paper in  Table 9 and 10 of Sec A.7.
>
>
> (c) **Impact of increasing the client participation with number of clients fixed**
>
> Unlike the previous ablation, here we fix the number of clients to 100 and vary the client participation rate from 5%, 10% and 15%. We consider the CIFAR-100 dataset with non-iid partitioning of Dirichlet ($\delta$=0.3). As expected, the accuracy of the FL-trained models improve with an increase in the client participation rate. We would also like to highlight here that, addition of our proposed ASD strategy consistently improves the accuracy when combined on top of the existing baseline algorithms such as FedAvg and FedDyn. These results are also reported in the revised manuscript (see Sec A.7, Table 11 )
>
>
> $\delta = 0.3$
>
> |            |       | client participation |       |
> |------------|-------|:--------------------:|-------|
> | Method     |   5%  |          10%         |  15%  |
> | FedAvg     | 38.22 |         38.67        | 38.85 |
> | FedAvg+ASD (ours) | **43.04** |         **42.77**        | **43.59** |
> | FedDyn     | 44.68 |         47.56        | 47.87 |
> | FedDyn+ASD (ours) | **47.51** |         **49.03**        | **50.32** |
>
> $\delta = 0.6$
>
> |            |           | client paticipation |           |
> |------------|-----------|:-------------------:|-----------|
> | Method     |     5%    |         10%         |    15%    |
> | FedAvg     |   39.04   |        38.53        |        38 |
> | FedAvg+ASD (Ours) | **43.51** |      **42.54**      |  **42.9** |
> | FedDyn     |   45.18   |        48.87        |     48.74 |
> | FedDyn+ASD (Ours) | **47.81** |      **51.44**      | **51.48** |

---

> ### Author Response · Authors · 2024-09-07
> **Response to Reviewer J9kY (3/3)**
>
> **(3) Performance when client model is Tiny-ViT with ImageNet-100 [2] dataset**
>
> We perform experiments with ViT architecture using the Tiny-ViT[1] as client models  on ImageNet-100 dataset [2] with non-iid data partitioning of Dirichlet $\delta = 0.3$. The choice of  Tiny-ViT is motivated by the fact that edge devices are traditionally computational resource-constrained and Tiny-ViT is designed for such applications. For this experiment, the number of clients is set to 200, and the client participation rate is set to 5\%. We have used 300 communication rounds.  In the Table below, we report the numbers averaged over 3 different trials. We observe that the addition of our proposed regularizer ASD atop FedAvg and FedDyn leads to consistent improvements, thereby further justifying the efficacy of our proposed method on the deeper architectures. These experiments are added  to the Sec.A.5 of the appendix in the paper.
>
>
> | Method     | Accuracy(\%) |
> |------------|--------------|
> | FedAvg     |         18.5 |
> | FedAvg+ASD |     **21.2** |
> | FedDyn     |        28.02 |
> | FedDyn+ASD |     **36.7** |
>
> [1]  TinyViT: Fast Pretraining Distillation for Small Vision Transformers, (ECCV, 2022)
>
> [2] DLME: Deep Local-flatness Manifold Embedding (ECCV,2022)
>
>
> **(4) Performance when client model is ViT-Small with CIFAR-100 dataset**
>
> We performed an experiment using ViT-Small architecture as client models on CIFAR-100. We observe that adding our ASD regularizer improves the baseline FedAvg by $1.4$\% and $1.7$\%  for $\delta = 0.3$ and $\delta = 0.6$, respectively. In this setup, we consider $100$ clients with 10\% participation, and the accuracy is reported at the end of $300$ rounds.
>
> | Method     | $\delta = 0.3$ | $\delta = 0.6$ |
> |------------|-------------|-------------|
> | FedAvg     |       53.22 |       52.63 |
> | FedAvg+ASD (Ours) |       **54.67** |       **54.34** |
>
> **(5) Optimization has always been the major problem. It is not entirely clear how is the presented algorithm is supposed to address problems with the optimization? What if the statistics of the server and client change over time, such that it is unclear when to switch from the global to the local model and vice versa?**
>
> We assume the training set on the clients to be fixed throughout training for a fair comparison with the existing SOTA methods as this has been an accepted experimental protocol in the heterogeneous FL literature such as [FedDyn], [FedNTD], etc.  Also, as mentioned in our very first response **(1)**, one of the main contributions of this work is the adaptive weighting scheme, which is based on client data distribution and automatically adjusts an adaptive weight to scale the regularization loss. In fact, our proposed ASD loss acts as a regularizer from the perspective of the client, i.e., each client minimizes the ASD loss along with the client-specific loss function, such as cross-entropy loss in the context of classification. We have shown that adding this loss reduces the Gradient Dissimilarity and thus helps in better convergence. This discussion was already done in section (A.12) of the original submission, and it is in Sec A.16. of the revised submission.
>
> [FedDyn] Federated Learning Based on Dynamic Regularization, ICLR,2021
>
> [FedNTD] Preservation of the Global Knowledge by Not-True Distillation in Federated Learning (NeurIPS,2022)

---

### Decision · Action_Editor_14WR · 2024-11-26

**Recommendation:** Accept with minor revision

**Comment:**

Three reviewers recommended acceptance. The authors have extensively experimented with the requested ablations on hyperparameters and experimental settings, including the value lambda, the size of the dataset overall as well as per different server-client ratios, temperature tau, different IID settings or ways of distributing labels, with larger datasets, and more modern and expressive architectures. They demonstrated sufficient generalization and performance improvement.

One reviewer pointed out that certain confusions remain in the introduction, where too many elements are convoluted. For non-experts in FL, this may present a clear barrier. Some suggestions in this respect include: clarifying in the title that the topic concerns class-imbalance federated learning; using Figure 1 to clearly introduce the problem and explain the proposed solution (it should be explicitly noted in the caption that the figure represents one round of local training on a specific client model; perhaps the figure can be resized and supplemented with illustrations involving several clients); briefly explaining FedAvg and emphasizing the truly plug-and-play nature of FedNTD + Ours, which should also be illustrated.
Additionally, the authors are encouraged to review the referencing style. For instance, the second sentence in the introduction should include proper bracketing, such as: many applications in smartphones **(**Hard et al. (2018); Ramaswamy et al. (2019)**)**, the Internet of Things (IoT), healthcare organizations **(**Rieke et al. (2020); Xu et al. (2021)**)**.

**Audience:**

Yes.

**Claims And Evidence:**

The manuscript addresses heterogeneous federated learning, where the empirical label distribution across clients varies. It modifies the optimization problem on the client side by adding a regularization term (the KL divergence between global and local models). For each sample, the weight assigned to the regularization loss is adaptively adjusted based on the global model’s prediction entropy and the empirical label distribution of the client’s data. The proposed method can be integrated with any existing FL approach. This is demonstrated by combining the proposed regularization scheme with several popular off-the-shelf FL methods, including FedAvg, FedProx, FedDyn, FedSpeed, FedNTD, FedSAM, and FedDisco. Experiments were conducted on CIFAR-10, CIFAR-100, and Tiny ImageNet, using both shallow architectures and more advanced models like ResNet-20 and Vision Transformer (ViT). Additionally, the manuscript provides a theoretical explanation for the reduction of client drift achieved through the proposed regularizer.